# LEARNING DEEP MODELS:
# CRITICAL POINTS AND LOCAL OPENNESS

## ABSTRACT

With the increasing interest in deeper understanding of the loss surface of many non-convex *deep models*, this paper presents a unifying framework to study the local/global optima equivalence of the optimization problems arising from training of such non-convex models. Using the *local openness* property of the underlying training models, we provide simple sufficient conditions under which any local optimum of the resulting optimization problem is globally optimal. We first *completely characterize the local openness of matrix multiplication mapping in its range*. Then we use our characterization to: 1) show that every local optimum of two layer linear networks is globally optimal. Unlike many existing results in the literature, our result requires no assumption on the target data matrix $\boldsymbol{Y}$, and input data matrix $\boldsymbol{X}$. 2) develop *almost complete* characterization of the local/global optima equivalence of multi-layer linear neural networks. We provide various counterexamples to show the necessity of each of our assumptions. 3) show global/local optima equivalence of non-linear deep models having certain pyramidal structure. Unlike some existing works, our result requires no assumption on the differentiability of the activation functions and can go beyond "full-rank" cases.

## 1   INTRODUCTION

Deep learning models have recently led to significant practical successes in various fields ranging from computer vision to natural language processing. Despite these significant empirical successes, the theoretical understanding of the behavior of these models is still very limited. While some recent works have tried to explain these successes through the lens of *expressivity* by showing the power of these models in learning large class of mappings, other works find the root of the success in the *generalizability* of these models from learning perspective.

From optimization perspective, training deep models require solving non-convex optimization problems, where non-convexity arises from the "deep" structure of the model. In fact, it has been shown by Blum & Rivest (1989) that training neural networks to global optimality is NP-complete in the worst case even for the simple case of three node networks. Despite this worst case barrier, the practical success of deep learning may suggest that most of the local optimal points of these models are close to the global optimal points. In particular, Choromanska et al. (2015) uses spin glass theory and empirical experiments to show that the local optima of deep neural network optimization problem are close to the global optima.

In an effort to better understand the landscape of training deep neural networks, Kawaguchi (2016); Lu & Kawaguchi (2017); Yun et al. (2017); Hardt & Ma (2016) studied the linear neural networks and provided sufficient conditions under which critical points (or local optimal points) of the training optimization problems are globally optimal. For non-linear neural networks, multiple works have shown that when the number of parameters of the model is larger than the data dimension, local optima of the resulting optimization problems can be easily found using local search procedures; see, e.g., Soltanolkotabi et al. (2017); Soudry & Carmon (2016); Nguyen & Hein (2017); Xie et al. (2017).

Despite the growing interest in studying the landscape of deep optimization problems, many of the results and mathematical analyses are problem specific and cannot be generalized to other problems and network structures easily. As a first step toward reaching a unifying theory for these results, we propose the use of open mappings for characterizing the properties of the local optima of these "deep" optimization problems.

To study the landscape of shallow/deep models, we study the general optimization problem

$$\underset{\boldsymbol{w} \in \mathcal{W}}{\text{minimize}} \; \ell(\mathcal{F}(\boldsymbol{w})), \tag{1}$$

where $\ell(\cdot)$ is the loss function and $\mathcal{F}(\cdot)$ represents a statistical model with parameter $\boldsymbol{w}$ that needs to be learned by solving the above optimization problem. A simple example is the popular linear regression problem

$$\underset{\boldsymbol{w}}{\text{minimize}} \, \|\boldsymbol{X}\boldsymbol{w} - \boldsymbol{y}\|_2^2,$$

where $\boldsymbol{y}$ is a given constant response vector and $\boldsymbol{X}$ is a given constant feature matrix. In this example, the loss function is the $\ell_2$ loss, i.e., $\ell(\boldsymbol{z}) = \|\boldsymbol{z} - \boldsymbol{y}\|_2^2$, and the fitted model $\mathcal{F}$ is a linear model, i.e., $\mathcal{F}(\boldsymbol{w}) = \boldsymbol{X}\boldsymbol{w}$. While this linear regression problem is convex and easy, fitting many practical models, such as deep neural networks, requires solving non-trivial non-convex optimization problems.

In this paper, we use the local openness of the mapping $\mathcal{F}$ to provide sufficient conditions under which every local optimum of (1) is in fact global optimum. To proceed, let us define our notations that will be used throughout the paper. We use $\boldsymbol{A}_{l,:}$ and $\boldsymbol{A}_{:,l}$ to denote the $l^{th}$ row and column of the matrix $\boldsymbol{A}$, respectively. The notation $\boldsymbol{I}_d \in \mathbb{R}^{d \times d}$ is used to denote the $d \times d$-dimensional identity matrix. Let $\|\boldsymbol{A}\|$, $\mathcal{N}(\boldsymbol{A})$, $\mathcal{C}(\boldsymbol{A})$, $\text{rank}(\boldsymbol{A})$ be respectively the Frobenius norm, null-space, column-space, and the rank of the matrix $\boldsymbol{A}$. Given subspaces $U$ and $V$, we say $U \perp V$ if $U$ is orthogonal to $V$, and $U = V^\perp$ if $U$ is the orthogonal complement of $V$. We say matrix $\boldsymbol{A} \in \mathbb{R}^{d_1 \times d_0}$ is rank *deficient* if $\text{rank}(\boldsymbol{A}) < \min\{d_1, d_0\}$, and *full rank* if $\text{rank}(\boldsymbol{A}) = \min\{d_1, d_0\}$. We call a point $\boldsymbol{W} = (\boldsymbol{W}_h, \dots, \boldsymbol{W}_1)$, with $\boldsymbol{W}_i \in \mathbb{R}^{d_i \times d_{i-1}}$, *non-degenerate* if $\text{rank}(\boldsymbol{W}_h \cdots \boldsymbol{W}_1) = \min_{0 \le i \le h} d_i$, and degenerate if $\text{rank}(\boldsymbol{W}_h \cdots \boldsymbol{W}_1) < \min_{0 \le i \le h} d_i$. We also say a point $\bar{\boldsymbol{W}}$ is a *second order saddle point* of an unconstrained optimization problem if the gradient of the objective function is zero at $\bar{\boldsymbol{W}}$ and the hessian of the objective function at $\bar{\boldsymbol{W}}$ has a negative eigenvalue. Let us start by briefly explaining the training problem of feedforward neural networks which will also be used as a motivation for our analysis:

**Example: Training Feedforward Neural Networks.** Consider the following multiple layer feedforward neural network optimization problem:

$$\underset{\boldsymbol{W}}{\text{minimize}} \, \frac{1}{2}\|\mathcal{F}_h(\boldsymbol{W}) - \boldsymbol{Y}\|^2$$

where $\mathcal{F}_h$ is defined in a recursive manner:

$$\mathcal{F}_k(\boldsymbol{W}) \triangleq \boldsymbol{\sigma}_k\big(\boldsymbol{W}_k \mathcal{F}_{k-1}(\boldsymbol{W})\big), \text{ for } k \in \{2, \dots, h\},$$

with

$$\mathcal{F}_1(\boldsymbol{W}) \triangleq \boldsymbol{\sigma}_1(\boldsymbol{W}_1 \boldsymbol{X}).$$

Here $h$ is the number of hidden units in our network; $\boldsymbol{\sigma}_k(\cdot)$ denotes the activation function of layer $k$; the matrix $\boldsymbol{W}_k \in \mathbb{R}^{d_k \times d_{k-1}}$ is the weight of layer $k$ with $\boldsymbol{W} \triangleq (\boldsymbol{W}_i)_{i=1}^h$ being our optimization variable. The matrix $\boldsymbol{X} \in \mathbb{R}^{d_0 \times n}$ is the input training data; and $\boldsymbol{Y} \in \mathbb{R}^{d_h \times n}$ is the target training data where $n$ is the number of samples; see, e.g. Goodfellow & Courville (2016). Notice that this problem is a special case of the optimization problem in (1) which can be obtained simply by setting our loss function to the $\ell_2$ loss, and setting $\mathcal{F} = \mathcal{F}_h$.

A special instance of this optimization problem was studied in Nguyen & Hein (2017) which considers the non-linear neural network with pyramidal structure (i.e. $d_i \le d_{i-1} \, \forall \, i = 1, \dots, h$ and $d_0 \ge n$). Note that this special network structure does not allow wide intermediate layers. (Nguyen & Hein, 2017, Theorem 3.8) shows that under some conditions, among which are the differentiability of the loss function $\ell(\cdot)$ and the activation function $\sigma(\cdot)$, if $\boldsymbol{W}$ is a critical point with $\boldsymbol{W}_i$'s being full row rank then it is a global minimum. In this paper, we will relax the differentiability assumption on both $\ell(\cdot)$ and $\sigma(\cdot)$; and we will show any local optimum is a global optimum of the objective function. Another special case is the linear feedforward network where the mapping $\sigma_k(\cdot)$ is the identity map in all layers, which leads to the optimization problem:

$$\underset{\boldsymbol{W}}{\text{minimize}} \, \frac{1}{2}\|\boldsymbol{W}_h \cdots \boldsymbol{W}_1 \boldsymbol{X} - \boldsymbol{Y}\|^2. \tag{2}$$

For this optimization problem, Lu & Kawaguchi (2017) showed that every local optimum of the objective function is globally optimal under some assumptions. More precisely, by using perturbation analysis, (Lu & Kawaguchi, 2017, Theorem 2.2) prove that when $\boldsymbol{X}$ and $\boldsymbol{Y}$ are full row rank, every local optimum in problem (2) is a local optimum of the following problem:

$$\begin{aligned} \underset{\boldsymbol{Z} \in \mathbb{R}^{d_h \times d_0}}{\text{minimum}} \quad & \frac{1}{2}\|\boldsymbol{Z}\boldsymbol{X} - \boldsymbol{Y}\|^2 \\ \text{subject to} \quad & \text{rank}(\boldsymbol{Z}) \le d_p \triangleq \min_{0 \le i \le h} d_i. \end{aligned} \tag{3}$$

Moreover, they show that when $\boldsymbol{X}$ is full row rank, every local optimum of problem (3) is a global optimum. Thus, with the sufficient condition that $\boldsymbol{X}$ and $\boldsymbol{Y}$ are both full row rank, every local optimum of problem (2) is a global

optimum. Another recent work Yun et al. (2017) shows the same result under similar set of assumptions. It is in fact not hard to see that one cannot relax the full rankness assumption of $\boldsymbol{Y}$ due to the following simple counterexample:

$$\boldsymbol{X} = \boldsymbol{I} \quad \boldsymbol{W}_3 = \begin{bmatrix} 1 \\ 0 \end{bmatrix}, \quad \boldsymbol{W}_2 = [\, 0 \,], \quad \boldsymbol{W}_1 = [\, 1 \quad 0 \,], \quad \boldsymbol{Y} = \begin{bmatrix} 0 & 0 \\ 0 & 1 \end{bmatrix}.$$

It is not hard to check that the point $\mathbf{W} = (\mathbf{W}_1, \mathbf{W}_2, \mathbf{W}_3)$ is a local optimum of a 3-layer deep linear model $\big($problem (2) with $h = 3\,\big)$ that is not a global optimum. However, we will show that if a given local optimum is non-degenerate (which is a simple checkable condition), the full rankness of $\boldsymbol{Y}$ can be relaxed. Moreover, for degenerate local optima, we show that if there exist $1 \leq p_1 < p_2 \leq h-1$ with $d_h > d_{p_2}$ and $d_0 > d_{p_1}$, we can find $\boldsymbol{Y}$ and $\boldsymbol{X}$ such that problem (2) has a local minimum that is not global. Otherwise, given any $\boldsymbol{X}$ and $\boldsymbol{Y}$, we present a method for constructing a descent direction from any given degenerate critical point that is not a global optimum; thus we show every degenerate local minimum is global.

**Other examples: Matrix Factorization and Matrix Completion.** In addition to the training of deep neural networks, the matrix completion problem also lies in the category of non-convex problems in (1). For the matrix completion problem, Park et al. (2016) shows that the non-convex matrix factorization formulation of the non-square matrix sensing problem has no spurious local optimum under restricted isometry property (RIP) conditions. Similar results were obtained for the symmetric matrix multiplication problem by Ge et al. (2016), and the non-convex factorized low-rank matrix recovery problem by Bhojanapalli et al. (2016). Like the analysis in Ge et al. (2016), we start with the fully observed matrix completion scenario:

$$\underset{\boldsymbol{W}_1 \in \mathbb{R}^{d_1 \times d_0}, \boldsymbol{W}_2 \in \mathbb{R}^{d_2 \times d_1}}{\text{minimize}} \frac{1}{2} \| \boldsymbol{W}_2 \boldsymbol{W}_1 - \boldsymbol{Y} \|^2. \tag{4}$$

This problem, which is also referred to as the low rank matrix estimation problem in Srebro & Jaakkola (2003), can also be viewed as a 2-layer linear neural network optimization problem with the input data matrix $\boldsymbol{X} = \boldsymbol{I}$. Clearly, this problem is much simpler than the general matrix completion problem and we only study it as a first step. Moreover, this optimization problem is a special case of (1) with the loss function being the $\ell_2$ loss, and the mapping $\mathcal{F}$ being defined as $\mathcal{F}(\boldsymbol{W}_1, \boldsymbol{W}_2) = \boldsymbol{W}_2 \boldsymbol{W}_1$. In this paper, using our framework, we show that every critical point of (4) is either a global minimum or a second-order saddle point. This result can be generalized to general loss function $\ell(\cdot)$ for degenerate critical points.

In addition to these results, one of our main contributions is the complete characterization of the local openness of the matrix multiplication mapping in its range. These results could be used in many other optimization problems for characterizing the local/global equivalence.

## 2  MATHEMATICAL FRAMEWORK

As discussed in the previous section, we are interested in solving

$$\underset{\boldsymbol{w} \in \mathcal{W}}{\text{minimize}} \ \ell(\mathcal{F}(\boldsymbol{w})), \tag{5}$$

where $\mathcal{F} : \mathcal{W} \mapsto \mathcal{S}$ is a mapping and $\ell : \mathcal{S} \mapsto \mathbb{R}$ is a loss function. Here we assume that the set $\mathcal{W}$ is closed and the mapping $\mathcal{F}$ is continuous. In non-convex scenarios, this optimization problem can only be solved up to "local optimality" by local search procedures; see Lee et al. (2016) for an example. To proceed, let us define the auxiliary optimization problem

$$\underset{\boldsymbol{s} \in \mathcal{S}}{\text{minimize}} \ \ell(\boldsymbol{s}), \tag{6}$$

where $\mathcal{S}$ is the range of the mapping $\mathcal{F}$. Since problem (6) minimizes the function $\ell(\cdot)$ over the range of the mapping $\mathcal{F}$, the global optimal objective values for problems (5) and (6) are the same. Moreover, there is a clear relation between the global optima of the two optimization problem through the mapping $\mathcal{F}$. However, the connection between the local optima of the two optimization problems is not clear. This connection, in particular, is important when the local optima of (6) are "nice" (e.g. globally optimal or close to optimal). In what follows, we establish the connection between the local optima of the optimization problems (5) and (6) under some simple sufficient conditions. This connection is then used to study the relation between local and global optima of (5) and (6) for various deep learning models. Let us first define the following concepts, which will help us state our simple sufficient condition.

- **Open mapping:** A mapping $\mathcal{F} : \mathcal{W} \to \mathcal{S}$ is said to be open, if for every open set $U \in \mathcal{W}$, $\mathcal{F}(U)$ is (relatively) open in $\mathcal{S}$.

- **Locally open mapping:** A mapping $\mathcal{F}(\cdot)$ is said to be locally open at $\boldsymbol{w}$ if for every $\epsilon > 0$, there exists $\delta > 0$ such that $\mathcal{B}_\delta\big(\mathcal{F}(\boldsymbol{w})\big) \subseteq \mathcal{F}\big(\mathcal{B}_\epsilon(\boldsymbol{w})\big)$. Here $\mathcal{B}_\delta(\boldsymbol{w}) \subseteq \mathcal{W}$ is an open ball with radius $\delta$ centered at $\boldsymbol{w}$, and $\mathcal{B}_\epsilon(\mathcal{F}(\boldsymbol{w})) \subseteq \mathcal{S}$ is the ball of radius $\epsilon$ centered at $\mathcal{F}(\boldsymbol{w})$.

By definition, openness of a mapping is stronger than local openness. Furthermore, it is not hard to see that a mapping is locally open everywhere if and only if it is open. A useful property of (locally) open mappings is that *the composition of two (locally) open maps is (locally) open.*

The following simple intuitive observation, which establishes a connection between the local optima of (5) and (6), is a major building block of our analyses.

**Observation 1.** *Suppose $\mathcal{F}(\cdot)$ is locally open at $\bar{\boldsymbol{w}}$. If $\bar{\boldsymbol{w}}$ is a local minimum of problem (5), then $\bar{\boldsymbol{s}} = \mathcal{F}(\bar{\boldsymbol{w}})$ is a local minimum of problem (6).*

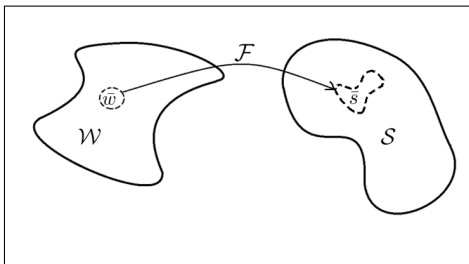

Figure 1: Sketch of the Proof of Observation 1.

*Proof.* Let $\bar{\boldsymbol{w}}$ be a local minimum of problem (5). Then there exists an $\epsilon > 0$ such that $\ell(\mathcal{F}(\bar{\boldsymbol{w}})) \leq \ell(\mathcal{F}(\boldsymbol{w})), \ \forall \boldsymbol{w} \in \mathcal{B}_\epsilon(\bar{\boldsymbol{w}})$. By the definition of local openness,

$$\exists \, \delta > 0 \text{ such that } \mathcal{B}_\delta(\bar{\boldsymbol{s}}) \subset \mathcal{F}\big(\mathcal{B}_\epsilon(\bar{\boldsymbol{w}})\big).$$

where $\bar{\boldsymbol{s}} = \mathcal{F}(\bar{\boldsymbol{w}})$. Therefore, $\ell(\bar{\boldsymbol{s}}) \leq \ell(\boldsymbol{s}), \ \forall \boldsymbol{s} \in \mathcal{B}_\delta(\bar{\boldsymbol{s}})$, which implies $\bar{\boldsymbol{s}}$ is a local minimum of problem (6). $\qquad\square$

The above observation can be used to map multiple local optima of the original problem (5) to one local optimum of the auxiliary problem (6); and potentially make the problem easier to understand. This mapping is particularly interesting in neural networks since permuting the weights in each layer does not change the objective function. Hence, by nature, the optimization problem has multiple (disconnected) global optima; and hence it is non-convex. However, collapsing these multiple local optima to one could potentially simplify the problem. In other words, instead of understanding the problem in the original variables, we can analyze it in the space of the resulted mapping. Let us clarify this point through the following simple examples:

**Example 2.** *Consider the optimization problem*

$$\underset{w \in \mathbb{R}}{minimize} \ (w^2 - 1)^2, \tag{7}$$

*and its corresponding auxiliary problem*

$$\underset{z \geq 0}{minimize} \ (z - 1)^2. \tag{8}$$

*Plots of these two problems can be found in Figure 2a and Figure 2b. Since $\mathcal{F}(x) \triangleq w^2$ is an open mapping in its range, it follows from Observation 1 that every local minimum in problem (7) is a local minimum of problem (8). Thus the two local minima $w = -1$ and $w = +1$ in (7) are mapped to a single local minimum $z = 1$ of problem (8). Moreover, since the optimization problem (8) is convex, the local minimum is global; and hence the original local optima $w = -1$ and $w = +1$ should be both global despite non-convexity of (7).*

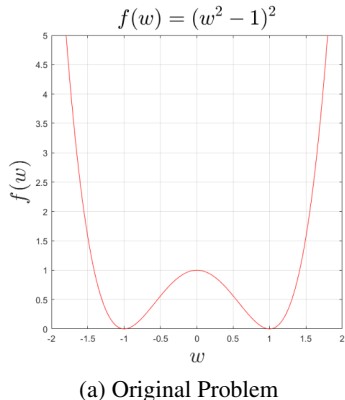
(a) Original Problem

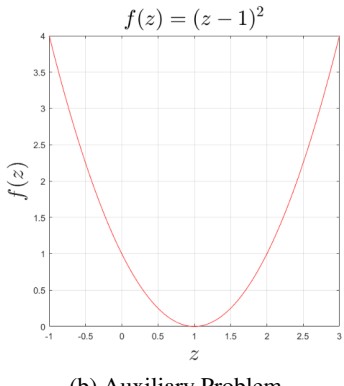
(b) Auxiliary Problem

Figure 2: Two local minima $w = -1$ and $w = +1$ in (a) are mapped to a single local minimum $z = 1$ in (b).

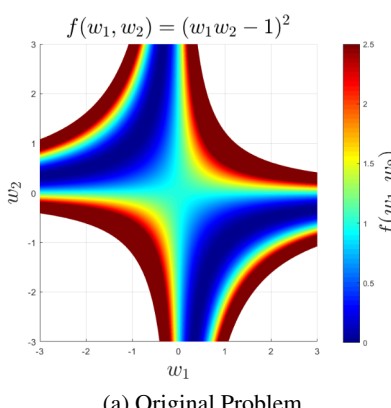
(a) Original Problem

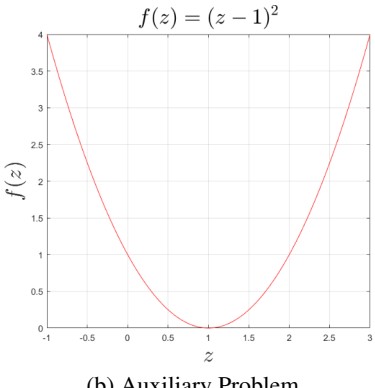
(b) Auxiliary Problem

Figure 3: All the points in the set $\{(w_1, w_2) \,|\, w_1 w_2 = 1\}$ are local minima in (a) and are mapped to a single local minimum $z = 1$ in (b).

**Example 3.** *Another example is related to the widely used matrix multiplication mapping $\boldsymbol{W}_1 \boldsymbol{W}_2$. Let $(\bar{\boldsymbol{W}}_1, \bar{\boldsymbol{W}}_2)$ be a local minimum of the optimization problem*

$$\underset{\boldsymbol{W}_1 \in \mathbb{R}^{m \times k},\, \boldsymbol{W}_2 \in \mathbb{R}^{k \times n}}{minimize} \ \ell(\boldsymbol{W}_1 \boldsymbol{W}_2).$$

*Then, any point in the set $\mathcal{S} \triangleq \{(\bar{\boldsymbol{W}}_1 \boldsymbol{Q}_1, \boldsymbol{Q}_2 \bar{\boldsymbol{W}}_2) \text{ with } \boldsymbol{Q}_1 \boldsymbol{Q}_2 = \boldsymbol{I}\}$ is also a local minimum. If the matrix product $\boldsymbol{W}_1 \boldsymbol{W}_2$ is locally open at the point $(\bar{\boldsymbol{W}}_1, \bar{\boldsymbol{W}}_2)$, then all points in $\mathcal{S}$ are mapped to a single local minimum $\boldsymbol{Z} = \boldsymbol{W}_1 \boldsymbol{W}_2$ in the corresponding auxiliary problem. A simple one dimensional example is plotted in Figure 3a and Figure 3b.*

This motivates us to study the local openness of the matrix multiplication mapping defined as

$$\mathcal{M} : \mathbb{R}^{m \times k} \times \mathbb{R}^{k \times n} \mapsto \mathcal{R}_{\mathcal{M}} \quad \text{with} \quad \mathcal{M}(\boldsymbol{W}_1, \boldsymbol{W}_2) \triangleq \boldsymbol{W}_1 \boldsymbol{W}_2, \tag{9}$$

where $\mathcal{R}_{\mathcal{M}} \triangleq \{\, \boldsymbol{Z} \in \mathbb{R}^{m \times n} \mid \text{rank}(\boldsymbol{Z}) \leq \min\{m, n, k\}\}$ is the range of the mapping $\mathcal{M}$.

Although matrix multiplication mappings $\mathcal{M}(\boldsymbol{W}_1, \boldsymbol{W}_2)$ naturally appears in deep models and is widely used as a non-convex factorization for rank constrained problems, see Wang et al. (2016); Bhojanapalli et al. (2016); Ge et al. (2016); Srebro & Jaakkola (2003); Sun (2015), to our knowledge, the complete characterization of the local openness of this mapping has not been studied in the optimization literature before.

While the classical open mapping theorem in Rudin (1973) states that surjective continuous linear operators are open, this is not true in general for bilinear mappings such as matrix product. In fact, by providing a simple counterexample

of a bilinear mapping that is not open, Horowitz (1975) shows that the linear case cannot be generally extended to multilinear maps. Several papers, see Balcerzak et al. (2013; 2005); Behrends (2011), investigate this bilinear mapping and provide a characterization of the points where this mapping is open. Moreover, Behrends (2017) studies the matrix multiplication mapping $\mathcal{M}$ which is a special example of bilinear mappings and provides an almost complete characterization of the points where the mapping is locally open. However, the openness is studied in $\mathbb{R}^{m \times n}$; while the range of the mapping is $\mathcal{R}_\mathcal{M}$; and the (relative) local openness should be studied with respect to this range in our framework. This, in particular causes trouble when $\mathbb{R}^{m \times n} \neq \mathcal{R}_\mathcal{M}$, i.e., when $k < \min\{m, n\}$.

For the above reason, we study the local openness of the mapping $\mathcal{M}$ in its range $\mathcal{R}_\mathcal{M}$ and characterize it completely. An intuitive (and unofficial) definition of local openness of $\mathcal{M}(\cdot)$ at $(\bar{\mathbf{W}}_1, \bar{\mathbf{W}}_2)$ in $\mathcal{R}_\mathcal{M}$ is as follows. We say the multiplication mapping is locally open at $(\bar{\mathbf{W}}_1, \bar{\mathbf{W}}_2)$ if for any small perturbation $\widetilde{\mathbf{Z}} \in \mathcal{R}_\mathcal{M}$ of $\bar{\mathbf{Z}} = \bar{\mathbf{W}}_1 \bar{\mathbf{W}}_2$, there exists a pair $(\widetilde{\mathbf{W}}_1, \widetilde{\mathbf{W}}_2)$, a small perturbation of $(\bar{\mathbf{W}}_1, \bar{\mathbf{W}}_2)$, such that $\widetilde{\mathbf{Z}} = \widetilde{\mathbf{W}}_1 \widetilde{\mathbf{W}}_2$.

Notice that when $k \geq \min\{m, n\}$, we get $\mathcal{R}_\mathcal{M} = \mathbb{R}^{m \times n}$. However, in the case where $k < \min\{m, n\}$ the mapping is definitely not locally open in $\mathbb{R}^{m \times n}$, but can still be locally open in $\mathcal{R}_\mathcal{M}$. As a simple example, consider $\bar{\mathbf{W}}_1 = \begin{bmatrix} 1 \\ 2 \end{bmatrix}$ and $\bar{\mathbf{W}}_2 = [\ 1 \quad 1\ ]$. In this example there does not exist $\widetilde{\mathbf{W}}_1, \widetilde{\mathbf{W}}_2$ perturbations of $\bar{\mathbf{W}}_1$ and $\bar{\mathbf{W}}_2$ respectively such that $\widetilde{\mathbf{W}}_1 \widetilde{\mathbf{W}}_2 = \widetilde{\mathbf{Z}}$ when $\widetilde{\mathbf{Z}}$ is a full rank perturbation of $\mathbf{Z} = \bar{\mathbf{W}}_1 \bar{\mathbf{W}}_2$; however, for any rank 1 perturbation $\widetilde{\mathbf{Z}}$, we can find a perturbed pair $(\widetilde{\mathbf{W}}_1, \widetilde{\mathbf{W}}_2)$ such that $\widetilde{\mathbf{Z}} = \widetilde{\mathbf{W}}_1 \widetilde{\mathbf{W}}_2$. Motivated by Observation 1, we study in the next section the local openness/openness of the mapping $\mathcal{M}$. We later use these results to analyze the behavior of local optima of deep neural networks.

## 3  LOCAL OPENNESS OF THE MATRIX MULTIPLICATION MAPPING

When $\boldsymbol{W}_1 \in \mathbb{R}^{m \times k}$ and $\boldsymbol{W}_2 \in \mathbb{R}^{k \times n}$ with $k \geq \min\{m, n\}$, the range of the mapping $\mathcal{M}(\boldsymbol{W}_1, \boldsymbol{W}_2) = \boldsymbol{W}_1 \boldsymbol{W}_2$ is the entire space $\mathbb{R}^{m \times n}$. In this case, which we refer to as the full rank case, (Behrends, 2017, Theorem 2.5) provides a complete characterization of the pairs $(\boldsymbol{W}_1, \boldsymbol{W}_2)$ for which the mapping is locally open. However, when $k < \min\{m, n\}$, which we refer to as the rank-deficient case, the characterization of the set of points for which the mapping is locally open has not been resolved before. We settled this question in Theorem 5 by providing a complete characterization of points $(\boldsymbol{W}_1, \boldsymbol{W}_2)$ for which the mapping $\mathcal{M}$ is locally open when $k < \min\{m, n\}$. We start by restating the main result in Behrends (2017):

**Proposition 4.** *(Behrends, 2017, Theorem 2.5 Rephrased) Let $\mathcal{M}(\mathbf{W}_1, \mathbf{W}_2) = \mathbf{W}_1 \mathbf{W}_2$ denote the matrix multiplication mapping with $\mathbf{W}_1 \in \mathbb{R}^{m \times k}$ and $\mathbf{W}_2 \in \mathbb{R}^{k \times n}$. Assume $k \geq \min\{m, n\}$. Then the the following statements are equivalent:*

*1. $\mathcal{M}(\cdot, \cdot)$ is locally open at $(\bar{\mathbf{W}}_1, \bar{\mathbf{W}}_2)$.*

*2.* $\begin{cases} \exists\, \widetilde{\mathbf{W}}_1 \in \mathbb{R}^{m \times k} \text{ such that } \widetilde{\mathbf{W}}_1 \bar{\mathbf{W}}_2 = \mathbf{0} \text{ and } \bar{\mathbf{W}}_1 + \widetilde{\mathbf{W}}_1 \text{ is full row rank.} \\ \qquad\qquad\qquad\qquad or \\ \exists\, \widetilde{\mathbf{W}}_2 \in \mathbb{R}^{k \times n} \text{ such that } \bar{\mathbf{W}}_1 \widetilde{\mathbf{W}}_2 = \mathbf{0} \text{ and } \bar{\mathbf{W}}_2 + \widetilde{\mathbf{W}}_2 \text{ is full column rank.} \end{cases}$

*3. $\dim\left(\mathcal{N}(\bar{\mathbf{W}}_1) \cap \mathcal{C}(\bar{\mathbf{W}}_2)\right) \leq k - m$  or  $n - (\text{rank}(\bar{\mathbf{W}}_2) - \dim\left(\mathcal{N}(\bar{\mathbf{W}}_1) \cap \mathcal{C}(\bar{\mathbf{W}}_2)\right) \leq k - \text{rank}(\bar{\mathbf{W}}_1)$.*

The above proposition provides a checkable condition which completely characterizes the local openness of the mapping $\mathcal{M}$ at different points when the range of the mapping is the entire space. Now, let us state our result that characterizes the local openness of the mapping $\mathcal{M}$ in its range when $k < \min\{m, n\}$.

**Theorem 5.** *Let $\mathcal{M}(\mathbf{W}_1, \mathbf{W}_2) = \mathbf{W}_1 \mathbf{W}_2$ denote the matrix multiplication mapping with $\mathbf{W}_1 \in \mathbb{R}^{m \times k}$ and $\mathbf{W}_2 \in \mathbb{R}^{k \times n}$. Assume $k < \min\{m, n\}$. Then if $\text{rank}(\bar{\mathbf{W}}_1) \neq \text{rank}(\bar{\mathbf{W}}_2)$, $\mathcal{M}(\cdot, \cdot)$ is not locally open at $(\bar{\mathbf{W}}_1, \bar{\mathbf{W}}_2)$. Else, if $\text{rank}(\bar{\mathbf{W}}_1) = \text{rank}(\bar{\mathbf{W}}_2)$, then the following statements are equivalent:*

*i) $\exists\, \widetilde{\mathbf{W}}_1 \in \mathbb{R}^{m \times k}$ such that $\widetilde{\mathbf{W}}_1 \bar{\mathbf{W}}_2 = \mathbf{0}$ and $\bar{\mathbf{W}}_1 + \widetilde{\mathbf{W}}_1$ is full column rank.*

*ii)* $\exists \, \widetilde{\mathbf{W}}_2 \in \mathbb{R}^{k \times n}$ *such that* $\bar{\mathbf{W}}_1 \widetilde{\mathbf{W}}_2 = \mathbf{0}$ *and* $\bar{\mathbf{W}}_2 + \widetilde{\mathbf{W}}_2$ *is full row rank.*

*iii)* $\dim \left( \mathcal{N}(\bar{\mathbf{W}}_1) \cap \mathcal{C}(\bar{\mathbf{W}}_2) \right) = 0.$

*iv)* $\dim \left( \mathcal{N}(\bar{\mathbf{W}}_2^T) \cap \mathcal{C}(\bar{\mathbf{W}}_1^T) \right) = 0.$

*v)* $\mathcal{M}(\cdot, \cdot)$ *is locally open at* $(\bar{\mathbf{W}}_1, \bar{\mathbf{W}}_2)$ *in its range* $\mathcal{R}_{\mathcal{M}}$.

Note that the proof of Theorem 5, which can be found in the appendix section, is different than the proof of Proposition 4, as in the former we need to work with the set of low rank matrices. Besides, the conditions in Theorem 5 are different than the ones in Proposition 4. For example, while conditions i) and ii) are equivalent in the rank-deficient case, they are not equivalent in the full-rank case. Moreover, unlike the full-rank case, the condition $\mathrm{rank}(\bar{\mathbf{W}}_1) = \mathrm{rank}(\bar{\mathbf{W}}_2)$ is necessary for local openness in the low rank case.

**How much perturbation is needed?** As previously mentioned, local openness can be described in terms of perturbation analysis. For example, $\mathcal{M}(\cdot, \cdot)$ is locally open at $(\boldsymbol{W}_1, \boldsymbol{W}_2)$ if for a given $\epsilon > 0$, there exists $\delta > 0$ such that for any $\tilde{\boldsymbol{Z}} = \boldsymbol{Z} + \boldsymbol{R}_\delta \in \mathcal{R}_{\mathcal{M}}$ with $\|\boldsymbol{R}_\delta\| \leq \delta$, there exists $\tilde{\boldsymbol{W}}_1, \tilde{\boldsymbol{W}}_2$ with $\|\tilde{\boldsymbol{W}}_1\| \leq \epsilon$, $\|\tilde{\boldsymbol{W}}_2\| \leq \epsilon$, such that $\tilde{\boldsymbol{Z}} = (\boldsymbol{W}_1 + \tilde{\boldsymbol{W}}_1)(\boldsymbol{W}_2 + \tilde{\boldsymbol{W}}_2)$. As a perturbation bound on $\delta$, we show that for any locally open pair $(\boldsymbol{W}_1, \boldsymbol{W}_2)$, given an $\epsilon > 0$, the chosen $\delta$ is of order $\epsilon$, i.e., $\delta = \mathcal{O}(\epsilon)$. The details of our analysis can be found in the proof of Theorem 5 in Appendix **B**.

**Remark 1** It follows from Theorem 5 that when $\boldsymbol{W}_1$ is full column rank, and $\boldsymbol{W}_2$ is full row rank, the mapping $\mathcal{M}(\cdot, \cdot)$ is locally open at $(\boldsymbol{W}_1, \boldsymbol{W}_2)$. This result was observed in other works; see, e.g., (Sun, 2015, Proposition 4.2). Also when $k < \min\{m, n\}$ if only one of the two matrices is full rank, then the mapping is not locally open. We have showed this result in the proof of Theorem 5, and below is a simple example for this phenomenon:

Let
$$\boldsymbol{W}_1 = \begin{bmatrix} 1 \\ 1 \end{bmatrix}, \quad \boldsymbol{W}_2 = [\,0\,,\,0\,], \quad \boldsymbol{W}_1 \boldsymbol{W}_2 = \begin{bmatrix} 0 & 0 \\ 0 & 0 \end{bmatrix}, \quad \boldsymbol{R}_\delta = \begin{bmatrix} \delta & 0 \\ 0 & 0 \end{bmatrix},$$

then $\boldsymbol{W}_1 \boldsymbol{W}_2 + \boldsymbol{R}_\delta$ is rank one and hence feasible perturbation. However, for any perturbation $\tilde{\boldsymbol{W}}_1 = \begin{bmatrix} \epsilon_1 \\ \epsilon_2 \end{bmatrix}$ and $\tilde{\boldsymbol{W}}_2 = [\,\epsilon_3\,,\,\epsilon_4\,]$, we have

$$(\boldsymbol{W}_1 + \tilde{\boldsymbol{W}}_1)(\boldsymbol{W}_2 + \tilde{\boldsymbol{W}}_2) = \begin{bmatrix} (1 + \epsilon_1)\epsilon_3 & (1 + \epsilon_1)\epsilon_4 \\ (1 + \epsilon_2)\epsilon_3 & (1 + \epsilon_2)\epsilon_4 \end{bmatrix}.$$

Hence, in order for this perturbation to be equal to $\boldsymbol{W}_1 \boldsymbol{W}_2 + \boldsymbol{R}_\delta$, we need $\epsilon_3$ to be different from zero. But when $\epsilon_3$ is different from zero, for small enough $\epsilon_2$, there does not exist such $\tilde{\boldsymbol{W}}_1$ and $\tilde{\boldsymbol{W}}_2$, or equivalently, $\mathcal{M}(\cdot, \cdot)$ is not locally open at $(\boldsymbol{W}_1, \boldsymbol{W}_2)$.

In the next sections, we use our local openness result to characterize the cases where the local optima of various training optimization problem of the form (5) are globally optimal.

## 4   NON-LINEAR DEEP NEURAL NETWORK WITH A PYRAMIDAL STRUCTURE:

Consider the non-linear deep neural network optimization problem with a pyramidal structure

$$\underset{\boldsymbol{W}}{\text{minimize}} \; \ell\big(\mathcal{F}_h(\boldsymbol{W})\big) \quad \text{with} \quad \mathcal{F}_1(\boldsymbol{W}) \triangleq \boldsymbol{\sigma}_1(\boldsymbol{W}_1 X); \quad \mathcal{F}_k(\boldsymbol{W}) \triangleq \boldsymbol{\sigma}_i\big(\boldsymbol{W}_i \mathcal{F}_{i-1}(\boldsymbol{W})\big), \tag{10}$$

for $i \in [2, h]$, where $\boldsymbol{\sigma}_i(\cdot)$ is the activation function applied component-wise to the entries of each layer, i.e., $\boldsymbol{\sigma}_i(\boldsymbol{A}) = [\sigma_i(\boldsymbol{A}_{ij})]_{i,j}$ with $\sigma_i : \mathbb{R} \mapsto \mathbb{R}$ being continuous and strictly monotone. Here $\boldsymbol{W} = \big(\boldsymbol{W}_i\big)_{i=1}^{h}$ where $\boldsymbol{W}_i \in \mathbb{R}^{d_i \times d_{i-1}}$ is the weight matrix of layer $i$, and $\boldsymbol{X} \in \mathbb{R}^{d_0 \times n}$ is the input training data. In this section, we consider the pyramidal network structure with $d_0 > n$ and $d_i \leq d_{i-1}$ for $1 \leq i \leq h$; see Nguyen & Hein (2017) for more details on these types of networks.

First notice that when $\boldsymbol{X}$ is full column rank and the functions $\sigma_i$'s are all continuous and strictly monotone, the image of the mapping $\mathcal{F}_h$ is convex and hence every local optimum of the auxiliary optimization problem (6) is global. We

now show that when $\boldsymbol{W}_i$'s are all full row rank and the functions $\sigma_i$ are all strictly monotone, the mapping $\mathcal{F}_h$ is locally open at $\boldsymbol{W}$.

**Lemma 6.** *Assume the functions $\sigma_i(\cdot) : \mathbb{R} \mapsto \mathbb{R}$ are all continuous strictly monotone. Then the mapping $\mathcal{F}_h$ defined in (10) is locally open at the point $\boldsymbol{W} = (\boldsymbol{W}_1, \ldots, \boldsymbol{W}_h)$ if $\boldsymbol{W}_i$'s are all full row rank.*

Before proving this result, we would like to remark that many of the popular activation functions such as logsitic, tangent hyperbolic, and leaky ReLu are strictly monotone and satisfy the assumptions of this lemma.

*Proof.* Let us prove by induction. Since linear mappings are open, and since $\sigma_1(\cdot)$ is strictly monotone; by using the composition property of open maps, we get that $\mathcal{F}_1$ is open.

Assume $\mathcal{F}_{k-1}\left(\left(\boldsymbol{W}_i\right)_{i=1}^{k-1}\right)$ is locally open at $\left(\boldsymbol{W}_i\right)_{i=1}^{k-1}$, then using Proposition 4, due to the full row rankness of $\boldsymbol{W}_k$, the mapping $\boldsymbol{W}_k \mathcal{F}_{k-1}\left(\left(\boldsymbol{W}_i\right)_{i=1}^{k-1}\right)$ is locally open at $\left(\boldsymbol{W}_k, \left(\boldsymbol{W}_i\right)_{i=1}^{k-1}\right)$. Using the composition property of open maps and strict monotonicity of $\sigma_k(\cdot)$, we get $\mathcal{F}_k\left(\left(\boldsymbol{W}_i\right)_{i=1}^{k}\right)$ is locally open at $\left(\boldsymbol{W}_i\right)_{i=1}^{k}$. □

Thus, by Observation 1, if $\bar{\mathbf{W}}$ is a local optimum of problem (10) with $\bar{\boldsymbol{W}}_i$'s being full row rank, then $\bar{\boldsymbol{Z}} = \mathcal{F}_h(\bar{\boldsymbol{W}})$ is a local optimum of the corresponding auxiliary problem:

$$\underset{\boldsymbol{Z} \in \mathcal{Z}}{\text{minimize }} \ell(\boldsymbol{Z})$$

where $\mathcal{Z}$ is convex. Consequently, $\bar{\boldsymbol{Z}}$ is a global optimum of problem (10) when the loss function $\ell(\cdot)$ is convex. Nguyen & Hein (2017) show that every critical point $\boldsymbol{W}$ of problem (10) with $\boldsymbol{W}_i$'s being full row rank is a global optimum when both $\sigma(\cdot)$ and $\ell(\cdot)$ are differentiable. Our result relaxes the differentiability assumption on both the activation and loss functions; however, we can only show all local optima are global. A popular activation function that is strictly monotonic and not differentiable is the Leaky ReLU, for which our result follows. It is also worth mentioning that Nguyen & Hein (2017) allow wide intermediate layers in parts of their result. It is not clear if this result can be extended to non-differentiable activation functions as well or not.

## 5  TWO-LAYER LINEAR NEURAL NETWORK

Consider the two layer linear neural network optimization problem

$$\underset{\boldsymbol{W}}{\text{minimize }} \frac{1}{2} \|\boldsymbol{W}_2 \boldsymbol{W}_1 \boldsymbol{X} - \boldsymbol{Y}\|^2 \tag{11}$$

where $\boldsymbol{W}_2 \in \mathbb{R}^{d_2 \times d_1}$ and $\boldsymbol{W}_1 \in \mathbb{R}^{d_1 \times d_0}$ are weight matrices, $\boldsymbol{X} \in \mathbb{R}^{d_0 \times n}$ is the input data, and $\boldsymbol{Y} \in \mathbb{R}^{d_2 \times n}$ is the target training data. Using our transformation, the corresponding auxiliary optimization problem can be written as

$$\begin{aligned} \underset{\boldsymbol{Z}}{\text{minimum}} \quad & \frac{1}{2} \|\boldsymbol{Z}\boldsymbol{X} - \boldsymbol{Y}\|^2 \\ \text{subject to} \quad & \text{rank}(\boldsymbol{Z}) \leq \min\{d_2, d_1, d_0\} \end{aligned} \tag{12}$$

(Lu & Kawaguchi, 2017, Theorem 2.2) shows that when $\boldsymbol{X}$ is full rank, every local minimum of problem (12) is global. By using local openness, we first show that this result holds without any assumption on $\boldsymbol{X}$ or $\boldsymbol{Y}$. The proof of Lemma 7 can be found in Appendix $A.3$

**Lemma 7.** *Every local minimum of problem (12) is global.*

Lemma 7 uses local openness to simplify the proof of (Lu & Kawaguchi, 2017, Theorem 2.2) and relax the full rankness assumption on $\boldsymbol{X}$. In another related work, (Kawaguchi, 2016, Theorem 2.3) shows that when $\boldsymbol{X}\boldsymbol{X}^T$ and $\boldsymbol{Y}\boldsymbol{X}^T$ are full rank, $d_2 \leq d_0$, and when $\boldsymbol{Y}\boldsymbol{X}^T(\boldsymbol{X}\boldsymbol{X}^T)^{-1}\boldsymbol{X}\boldsymbol{Y}^T$ has $d_2$ distinct eigenvalues, every local optimum is global and all saddle points are second order saddles. While the local/global equivalence result holds for deeper networks, the property that all saddles are second order does not hold in that case. Another result by (Yun et al., 2017, Theorem 2.2) shows that when $\boldsymbol{X}\boldsymbol{X}^T$, $\boldsymbol{Y}\boldsymbol{X}^T$, and $\boldsymbol{Y}\boldsymbol{X}^T(\boldsymbol{X}\boldsymbol{X}^T)^{-1}\boldsymbol{X}\boldsymbol{Y}^T$ are full rank, every local optimum of a linear deep network is global. Moreover, they provide necessary and sufficient conditions for a critical point to be a

global minimum. However, in their proof, the full rankness assumption of $\boldsymbol{Y}\boldsymbol{X}^T$ was not used in showing the result for non-degenerate critical points and thus can be relaxed in that case. In this section, without any assumptions on both $\boldsymbol{X}$ and $\boldsymbol{Y}$, we reconstruct the proof that shows the latter result for 2-layer networks using local openness, and then show a similar result for the degenerate case. The result for the degenerate case holds when replacing the square loss error by a general convex loss function as we will see in Colorollary 9. The proofs of the theorem and corollary stated below can be found in Appendices $A.1$ and $A.2$, respectively.

**Theorem 8.** *Every local minimum of problem (11) is global. Moreover, every degenerate saddle point of problem (11) is a second order saddle.*

**Corollary 9.** *Let the square loss error in (11) be replaced by a general convex loss function $\ell(\cdot)$. Then every degenerate critical point is either a global minimum or a second order saddle.*

Baldi & Hornik (1989) and Srebro & Jaakkola (2003) show the same result when both $\boldsymbol{X}$ and $\boldsymbol{Y}$ are full row rank. Theorem 8 generalizes their results by relaxing the assumptions on both $\boldsymbol{X}$ and $\boldsymbol{Y}$.

## 6   Multi-Layer Linear Neural Network

Consider the training problem of multi-layer deep linear neural networks:

$$\underset{\boldsymbol{W}}{\text{minimize}} \ \frac{1}{2}\|\boldsymbol{W}_h\cdots\boldsymbol{W}_1\boldsymbol{X} - \boldsymbol{Y}\|^2. \tag{13}$$

Here $\boldsymbol{W} = \left(\boldsymbol{W}_i\right)_{i=1}^h$, $\boldsymbol{W}_i \in \mathbb{R}^{d_i \times d_{i-1}}$ are the weight matrices, $\boldsymbol{X} \in \mathbb{R}^{d_0 \times n}$ is the input training data, and $\boldsymbol{Y} \in \mathbb{R}^{d_h \times n}$ is the target training data. Based on our general framework, the corresponding auxiliary optimization problem is given by

$$\begin{aligned} \underset{\boldsymbol{Z}\in\mathbb{R}^{d_h\times n}}{\text{minimum}} \quad & \frac{1}{2}\|\boldsymbol{Z}\boldsymbol{X} - \boldsymbol{Y}\|^2 \\ \text{subject to} \quad & \text{rank}(\boldsymbol{Z}) \le d_p \triangleq \min_{0\le i\le h} d_i \end{aligned} \tag{14}$$

Paper Lu & Kawaguchi (2017) showed that when $\boldsymbol{X}$ and $\boldsymbol{Y}$ are full row rank, every local minimum of (13) is global. We now relax the full rankness assumption and reproduce similar results. However, as we will see, the local/global equivalence does not always follow if we relax the full rankness. In such cases, we will provide detailed counter examples. Before proceeding to the proof we define the following mapping:

$$\mathcal{M}_{i,j}(\boldsymbol{W}_i,\ldots,\boldsymbol{W}_j):\{\boldsymbol{W}_i,\ldots,\boldsymbol{W}_j\}\to\mathcal{R}_{\mathcal{M}_{i,j}}\triangleq\{\boldsymbol{Z}=\boldsymbol{W}_i\ldots\boldsymbol{W}_j\in\mathbb{R}^{d_i\times d_{j-1}}\,|\,\text{rank}(\boldsymbol{Z})\le\min_{j-1\le l\le i}d_l\} \quad \text{for } i > j$$

Now we state Theorem 3.1 of Lu & Kawaguchi (2017) using our notation. The proof of the lemma stated below can be found in Appendix $A.4$.

**Lemma 10.** *If $\boldsymbol{W}$ is non-degenerate, then $\mathcal{M}_{h,1}(\boldsymbol{W}) = \boldsymbol{W}_h\cdots\boldsymbol{W}_1$ is locally open at $\boldsymbol{W}$.*

We now demonstrate our main results for this optimization problem which shows that under a set of necessary conditions, every local minimum of problem (13) is global. Although the result for the non-degenerate case directly follows from (Yun et al., 2017, Theorem 2.2), we provide in Lemma 11 a more intuitive proof that uses local openness of $\mathcal{M}$. Moreover, Theorem 12 extends the result to degenerate critical points.

**Lemma 11.** *Every non-degenerate local minimum of (13) is global minimum.*

*Proof.* Suppose $\boldsymbol{W} = (\boldsymbol{W}_h,\ldots,\boldsymbol{W}_1)$ is a non-degenerate local minimum. Then it follows by Lemma 10 that $\mathcal{M}_{h,1}$ is locally open at $\boldsymbol{W}$. Then by Lemma 1, $\boldsymbol{Z} = \mathcal{M}_h(\boldsymbol{W}_h,\ldots,\boldsymbol{W}_1)$ is a local optimum of problem (14) which is in fact global by Lemma 7. $\qquad\square$

We now state the desired result for degenerate critical points.

**Theorem 12.** *Let $p_1^* \triangleq \underset{0\le i\le h}{\text{argmin}}\ d_i$ and $p_2^* \triangleq \underset{j\ne p_1^*}{\text{argmin}}\ d_j$. If $d_{p2*} < min\{d_h, d_0\}$, we can find a rank deficient $\boldsymbol{Y}$ such that problem (30) has a local minimum that is not global. Otherwise, given any $\boldsymbol{X}$ and $\boldsymbol{Y}$, every local minimum of problem (30) is a global minimum.*

The proof of Theorem 12 can be found in Appendix $C$.

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

# Appendix

## A    PROOF OF THEOREM 8, COROLLARY 9 AND LEMMA 10

### A.1    PROOF OF THE THEOREM 8

*Proof.* The proof for the degenerate case is done by constructing a descent direction if the point is critical but not global. Let $(\bar{W}_2, \bar{W}_1)$ be a degenerate critical point, i.e. $\text{rank}(\bar{W}_2\bar{W}_1) < \min\{d_2, d_1, d_0\}$. Then, based on the dimensions of $d_0$, $d_1$, and $d_2$, we have one of the following cases:

$d_2 < d_1$ then $\exists\, \boldsymbol{b} \neq \mathbf{0}$ such that $\boldsymbol{b} \in \mathcal{N}(\bar{W}_2)$

$d_0 < d_1$ then $\exists\, \boldsymbol{b} \neq \mathbf{0}$ such that $\boldsymbol{b} \in \mathcal{N}(\bar{W}_1^T)$

$d_1 \leq d_2$, and $d_1 \leq d_0$ then either $\bar{W}_2$ is rank deficient and $\exists\, \boldsymbol{b} \neq \mathbf{0}$ such that $\boldsymbol{b} \in \mathcal{N}(\bar{W}_2)$ or

$\bar{W}_1$ is rank deficient and $\exists\, \boldsymbol{b} \neq \mathbf{0}$ such that $\boldsymbol{b} \in \mathcal{N}(\bar{W}_1^T)$

So in all cases either $\mathcal{N}(\bar{W}_2) \neq \emptyset$ or $\mathcal{N}(\bar{W}_1^T) \neq \emptyset$. Also, let $\boldsymbol{\Delta} = \bar{W}_2\bar{W}_1\boldsymbol{X} - \boldsymbol{Y}$. If $\boldsymbol{\Delta}\boldsymbol{X}^T = \mathbf{0}$, then by convexity of the square loss error function, the point $(\bar{W}_2, \bar{W}_1)$ is a global minimum of (11). Else, there exists $(i, j)$ such that $\langle \boldsymbol{X}_{i,:}, \boldsymbol{\Delta}_{j,:} \rangle \neq 0$. We now use first and second order optimality conditions to construct a descent direction when the current critical point is not global.

**First order optimality condition:** By considering perturbation in the directions $\boldsymbol{A} \in \mathbb{R}^{d_2 \times d_1}$ and $\boldsymbol{B} \in \mathbb{R}^{d_1 \times d_0}$ for the optimization problem

$$\underset{t}{\text{minimize}} \; \frac{1}{2}\|(W_2 + t\boldsymbol{A})(W_1 + t\boldsymbol{B})\boldsymbol{X} - \boldsymbol{Y}\|^2 \tag{15}$$

we obtain

$$\left\langle \boldsymbol{A}\bar{W}_1\boldsymbol{X} + \bar{W}_2\boldsymbol{B}\boldsymbol{X}, \boldsymbol{\Delta} \right\rangle = 0, \quad \forall \boldsymbol{A} \in \mathbb{R}^{d_2 \times d_1}, \boldsymbol{B} \in \mathbb{R}^{d_1 \times d_0}$$

**Second order optimality condition:**

$$2\left\langle \boldsymbol{A}\boldsymbol{B}\boldsymbol{X}, \boldsymbol{\Delta} \right\rangle + \|\boldsymbol{A}\bar{W}_1\boldsymbol{X} + \bar{W}_2\boldsymbol{B}\boldsymbol{X}\|^2 \geq 0 \quad \forall \boldsymbol{A} \in \mathbb{R}^{d_2 \times d_1}, \boldsymbol{B} \in \mathbb{R}^{d_1 \times d_0}$$

Suppose $(\bar{\mathbf{W}}_2, \bar{\mathbf{W}}_1)$ is a critical point and there exists $\boldsymbol{b} \neq 0$, $\boldsymbol{b} \in \mathcal{N}(\bar{\mathbf{W}}_2)$. We define

$$\boldsymbol{B}_{:,l} \triangleq \begin{cases} \alpha\boldsymbol{b} & \text{if } l = i, \\ \mathbf{0} & \text{otherwise} \end{cases} \qquad \boldsymbol{A}_{l,:} \triangleq \begin{cases} \boldsymbol{b}^T & \text{if } l = j, \\ 0 & \text{otherwise} \end{cases}$$

where $\alpha$ is a scalar constant. Then, using the second order optimality condition, for $\boldsymbol{c} = \|\boldsymbol{A}\bar{\mathbf{W}}_1\boldsymbol{X}\|^2$, we get

$$\alpha \underbrace{\|\boldsymbol{b}\|^2}_{\neq 0} \underbrace{\left\langle \boldsymbol{X}_{i,:}, \boldsymbol{\Delta}_{j,:} \right\rangle}_{\neq 0} + c \geq 0$$

Since this is true for every value of $\alpha$, $\boldsymbol{b}$ should be zero which contradicts the assumption on the choice of $\boldsymbol{b}$. Hence $\mathcal{N}(\bar{\mathbf{W}}_2) = \emptyset$.

Similarly, suppose $(\bar{\mathbf{W}}_2, \bar{\mathbf{W}}_1)$ is a critical point and there exists $\boldsymbol{a}^T \neq 0$, $\boldsymbol{a}^T \in \mathcal{N}(\bar{\mathbf{W}}_1^T)$. Let

$$\boldsymbol{A}_{l,:} \triangleq \begin{cases} \alpha\boldsymbol{a}^T & \text{if } l = j, \\ \mathbf{0} & \text{otherwise} \end{cases} \qquad \boldsymbol{B}_{:,l} \triangleq \begin{cases} \boldsymbol{a} & \text{if } l = i, \\ \mathbf{0} & \text{otherwise} \end{cases}$$

where $\alpha$ is a constant. Then, for $\boldsymbol{c} = \|\bar{\mathbf{W}}_2\boldsymbol{B}\boldsymbol{X}\|^2$, we get

$$\alpha \underbrace{\|\boldsymbol{a}\|^2}_{\neq 0} \underbrace{\left\langle \boldsymbol{X}_{i,:}, \boldsymbol{\Delta}_{j,:} \right\rangle}_{\neq 0} + c \geq 0$$

Using the same argument, we can show that $(\bar{\mathbf{W}}_2, \bar{\mathbf{W}}_1)$ is a second order saddle point of problem (11).

We now show the result for the non-degenerate case. Let $(\bar{\mathbf{W}}_2, \bar{\mathbf{W}}_1)$ be a non-degenerate local minimum, i.e. $\text{rank}(\bar{\mathbf{W}}_2\bar{\mathbf{W}}_1) = \min\{d_2, d_1, d_0\}$. Then it follows by Lemma 10 that the matrix multiplication $\mathcal{M}(\cdot, \cdot)$ is locally open at $(\bar{\mathbf{W}}_2, \bar{\mathbf{W}}_1)$. Then by Observation 1, $\boldsymbol{Z} = \bar{\mathbf{W}}_2\bar{\mathbf{W}}_1$ is a local optimum of problem (12) which is in fact global by Lemma 7.

$\square$

## A.2 PROOF OF COROLLARY 9

*Proof.* We follow the same steps used in the proof of Theorem 8 to show the result.

**First order optimality condition:** By considering perturbation in the directions $A \in \mathbb{R}^{d_2 \times d_1}$ and $B \in \mathbb{R}^{d_1 \times d_0}$ for the optimization problem

$$\underset{t}{\text{minimize}} \; \ell\big((W_2 + tA)(W_1 + tB)X - Y\big) \tag{16}$$

we obtain

$$\langle A\bar{W}_1 X + \bar{W}_2 B X, \nabla\ell(\bar{W}_2\bar{W}_1 X - Y)\rangle = 0 \quad \forall A \in \mathbb{R}^{d_2 \times d_1},\ B \in \mathbb{R}^{d_1 \times d_0}$$

Second order optimality condition:

$$2\langle ABX,\ \nabla\ell(\bar{W}_2\bar{W}_1 X - Y)\rangle + h\big(A\bar{W}_1 X,\ \bar{W}_2 B X,\ \bar{W}_2\bar{W}_1 X\big) \geq 0 \;\; \forall A \in \mathbb{R}^{d_2 \times d_1},\ B \in \mathbb{R}^{d_1 \times d_0}$$

where $h(\cdot)$ is a function that has a tensor representation. But we only need to know that it is a function of $A\bar{W}_1 X$, $\bar{W}_2 B X$, and $\bar{W}_2\bar{W}_1 X$.

If $\nabla\ell(\bar{W}_2\bar{W}_1 X - Y)X^T$, then by convexity of $\ell(\cdot)$, $(\bar{W}_2, \bar{W}_1)$ is a global minimum. Otherwise, there exists $(i,j)$ such that $\langle X_{i,:}, (\nabla\ell(\bar{W}_2\bar{W}_1 X - Y))_{j,:}\rangle \neq 0$. Using the same former argument in proof of Theorem 8, we choose $A$ and $B$ such that $h(A\bar{W}_1 X, \bar{W}_2 B X, \bar{W}_2\bar{W}_1 X)$ is some constant that does not depend on $\alpha$, and $\langle ABX, \nabla\ell(\bar{W}_2\bar{W}_1 X - Y)\rangle = \alpha \underbrace{\langle X_{i,:}, (\nabla\ell(\bar{W}_2\bar{W}_1 X - Y))_{j,:}\rangle}_{\neq 0} \neq 0$. Then by proper choice of $\alpha$ we show

that the point $(\bar{W}_2, \bar{W}_1)$ is a second order saddle point.

$\square$

## A.3 PROOF OF LEMMA 7

*Proof.* Let $r_X = \text{rank}(X)$ and $U_X \Sigma_X V_X^T$ with $U_X \in \mathbb{R}^{d_0 \times d_0}$, $\Sigma_X \in \mathbb{R}^{d_0 \times n}$, $V_X \in \mathbb{R}^{n \times n}$ be a singular value decomposition of $X$. Then

$$\|ZX - Y\|^2 = \left\|ZU_X \left[(\Sigma_X)_{:,1:r_X} | 0\right] - YV_X\right\|^2$$
$$= \|ZU_X(\Sigma_X)_{:,1:r_X} - (YV_X)_{:,1:r_X}\|^2 + \underbrace{\|(YV_X)_{:,r_X+1:n}\|^2}_{\text{constant in problem (12)}}.$$

Since $U_X(\Sigma_X)_{:,1:r_X}$ is full column rank, then the linear mapping $ZU_X(\Sigma_X)_{:,1:r_X}$ is open, and

$$\text{rank}(ZU_X(\Sigma_X)_{:,1:r_X}) \leq min\{\text{rank}(Z), r_X\} = min\{d_2, d_1, d_0, r_X\}.$$

Consequently, every local minimum in problem (12) corresponds to a local minimum in problem

$$\begin{array}{ll} \underset{\bar{Z} \in \mathbb{R}^{d_2 \times r_X}}{\text{minimum}} & \dfrac{1}{2}\|\bar{Z} - \bar{Y}\|^2 \\ \text{subject to} & \text{rank}(\bar{Z}) \leq \min\{d_2, d_1, d_0, r_X\} \end{array} \tag{17}$$

where $\bar{Y} = (YV_X)_{:,1:r_X}$. The result follows using (Lu & Kawaguchi, 2017, Theorem 2.2).

$\square$

## A.4 PROOF OF LEMMA 10

*Proof.* We construct a proof by induction on $h$ to show the desired result. When $h = 2$, we either have $d_1 < \min\{d_2, d_0\}$ or $d_1 \geq \min\{d_2, d_0\}$. In the first case,

$$d_1 = \text{rank}(W_2 W_1) \leq \text{rank}(W_1) \leq d_1 \Leftrightarrow \text{rank}(W_1) = d_1, \quad \text{and} \quad d_1 = \text{rank}(W_2 W_1) \leq \text{rank}(W_2) \leq d_1 \Leftrightarrow \text{rank}(W_2) = d_1.$$

Since $\boldsymbol{W}_1$ is full column rank and $\boldsymbol{W}_2$ is full column rank, then by Theorem 5, $\mathcal{M}_{2,1}(\cdot)$ is locally open at $(\boldsymbol{W}_2, \boldsymbol{W}_1)$. In the second case, either

$$d_2 = \mathrm{rank}(\boldsymbol{W}_2\boldsymbol{W}_1) \leq \mathrm{rank}(\boldsymbol{W}_2) \leq d_2 \Leftrightarrow \mathrm{rank}(\boldsymbol{W}_2) = d_2,$$

or

$$d_0 = \mathrm{rank}(\boldsymbol{W}_2\boldsymbol{W}_1) \leq \mathrm{rank}(\boldsymbol{W}_1) \leq d_0 \Leftrightarrow \mathrm{rank}(\boldsymbol{W}_1) = d_0$$

Thus, either $\boldsymbol{W}_2$ is full row rank or $\boldsymbol{W}_1$ is full row rank, then by Proposition 4, $\mathcal{M}_{2,1}(\cdot)$ is locally open at $(\boldsymbol{W}_2, \boldsymbol{W}_1)$.

Now assume the result holds for the product of $h$ matrices $\mathcal{M}_{h,1}(\boldsymbol{W})$, we show it is true for $\mathcal{M}_{h+1,1}(\boldsymbol{W})$.

Since
$$d_p = \mathrm{rank}(\boldsymbol{W}_h \ldots \boldsymbol{W}_1) \leq \mathrm{rank}(\boldsymbol{W}_{p+1}\boldsymbol{W}_p) \leq d_p \Leftrightarrow \mathrm{rank}(\boldsymbol{W}_{p+1}\boldsymbol{W}_p) = d_p,$$

then using Proposition 4, we get $\mathcal{M}_{p+1,p}(\cdot)$ is locally open at $(\boldsymbol{W}_{p+1}, \boldsymbol{W}_p)$. So we can replace $\boldsymbol{W}_{p+1}\boldsymbol{W}_p$ by a new matrix $\boldsymbol{Z}_p$ with rank $d_p$. Then by induction hypothesis, the product mapping $\mathcal{M}_{h+1,1} = \boldsymbol{W}_{h+1} \cdots \boldsymbol{W}_{p+2}\boldsymbol{Z}_p\boldsymbol{W}_{p-1} \cdots \boldsymbol{W}_1$ is locally open at $\boldsymbol{W}$. Since the composition of locally open maps is locally open, the result follows.

$\square$

## B  PROOF OF THEOREM 5

In this section, we prove Theorem 5. This theorem provides a complete characterization of points $(\bar{\mathbf{W}}_1, \bar{\mathbf{W}}_2) \in \mathbb{R}^{m \times k} \times \mathbb{R}^{k \times n}$ for which the matrix multiplication mapping $\mathcal{M}(\cdot, \cdot)$ is locally open for the case of $k < \min\{m, n\}$. In particular, we show that if $\mathrm{rank}(\bar{\mathbf{W}}_1) \neq \mathrm{rank}(\bar{\mathbf{W}}_2)$, $\mathcal{M}(\cdot, \cdot)$ is not locally open at $(\bar{\mathbf{W}}_1, \bar{\mathbf{W}}_2)$. Else, if $\mathrm{rank}(\bar{\mathbf{W}}_1) = \mathrm{rank}(\bar{\mathbf{W}}_2)$, then the following statements are equivalent:

i) $\exists \widetilde{\mathbf{W}}_1 \in \mathbb{R}^{m \times k}$ such that $\widetilde{\mathbf{W}}_1\bar{\mathbf{W}}_2 = \mathbf{0}$ and $\bar{\mathbf{W}}_1 + \widetilde{\mathbf{W}}_1$ is full column rank.

ii) $\exists \widetilde{\mathbf{W}}_2 \in \mathbb{R}^{k \times n}$ such that $\bar{\mathbf{W}}_1\widetilde{\mathbf{W}}_2 = \mathbf{0}$ and $\bar{\mathbf{W}}_2 + \widetilde{\mathbf{W}}_2$ is full row rank.

iii) $\dim \left( \mathcal{N}(\bar{\mathbf{W}}_1) \cap \mathcal{C}(\bar{\mathbf{W}}_2) \right) = 0$.

iv) $\dim \left( \mathcal{N}(\bar{\mathbf{W}}_2^T) \cap \mathcal{C}(\bar{\mathbf{W}}_1^T) \right) = 0$.

v) $\mathcal{M}(\cdot, \cdot)$ is locally open at $(\bar{\mathbf{W}}_1, \bar{\mathbf{W}}_2)$ in its range $\mathcal{R}_{\mathcal{M}} = \{\boldsymbol{Z} \in \mathbb{R}^{m \times n} \text{ with } \mathrm{rank}(\boldsymbol{Z}) \leq \min\{m, k, n\}\}$.

To prove this result, we first show that the local openness of $\mathcal{M}(\cdot, \cdot)$ at $(\boldsymbol{W}_1, \boldsymbol{W}_2)$ is equivalent to the local openness of $\mathcal{M}(\cdot, \cdot)$ at $(\boldsymbol{U}^T\boldsymbol{W}_1, \boldsymbol{W}_2\boldsymbol{V})$ where the columns of $\boldsymbol{U} \in \mathbb{R}^{m \times m}$ and the columns of $\boldsymbol{V} \in \mathbb{R}^{n \times n}$ are the left and right singular vectors of the product $\boldsymbol{W}_1\boldsymbol{W}_2$, respectively. This allows us to focus our study on the local openness of the mapping $\mathcal{M}$ to matrix pairs whose product is a diagonal matrix. We then show in Lemma 16 that when $\mathrm{rank}(\boldsymbol{W}_1) = \mathrm{rank}(\boldsymbol{W}_2)$, the statements $i, ii, ii$, and $iv$ are equivalent. Finally, we show in Proposition 18, that these conditions hold if and only if the mapping $\mathcal{M}(\cdot, \cdot)$ is locally open at $(\boldsymbol{W}_1, \boldsymbol{W}_2)$. Before proceeding we state and prove Lemma 13 that will be used later in the proof.

**Lemma 13.** *Let $\boldsymbol{V} \in \mathbb{R}^{m \times n}$ be a matrix with $\mathrm{rank}(\boldsymbol{V}) = r < m$. Then there exist an index set $\mathcal{B} = \{i_1, \ldots, i_r\} \subseteq \{1, \ldots, m\}$ and a matrix $\boldsymbol{A} \in \mathbb{R}^{(m-r) \times r}$ such that*

$$\|\boldsymbol{A}\|_\infty = \max_{i,j} |\boldsymbol{A}_{ij}| \leq 2^{m-r-1} \quad and \quad \boldsymbol{V}_{\mathcal{B}^c} = \boldsymbol{A}\boldsymbol{V}_{\mathcal{B}},$$

*where $\boldsymbol{V}_{\mathcal{B}} \in \mathbb{R}^{r \times n}$ is a matrix with rows $\{\boldsymbol{V}_{i,:}\}_{i \in \mathcal{B}}$ and $\boldsymbol{V}_{\mathcal{B}^c} \in \mathbb{R}^{(m-r) \times n}$ is a matrix with rows $\{\boldsymbol{V}_{i,:}\}_{i \in \mathcal{B}^c}$.*

Notice that in the above lemma, the bound on the norm of matrix $\mathbf{A}$ is independent of the dimension $n$ and it also does not depend on the choice of matrix $\mathbf{V}$.

*Proof.* To ease the notation, we denote the $i^{th}$ row of $\boldsymbol{V}$ by $\boldsymbol{v}_i$. We use induction on $m$ to show that there exists a basis $\mathcal{B} = \{i_1, \ldots, i_r\}$ and a vector $\boldsymbol{a}_j \in \mathbb{R}^r$ such that $\forall\, j \in \mathcal{B}^c$,

$$\boldsymbol{v}_j = \sum_{i \in \mathcal{B}} \boldsymbol{a}_{j,i} \boldsymbol{v}_i \qquad \text{with } |\boldsymbol{a}_{j,i}| \le 2^{m-r-1} \quad \forall\, i \in \mathcal{B}.$$

• *Induction Base Case* $m = r + 1$: Without loss of generality, assume $\mathcal{B} = \{1, \ldots, r\}$. Since the case of $\boldsymbol{v}_{r+1} = 0$ trivially holds, we consider $\boldsymbol{v}_{r+1} \ne 0$. By the property of basis, there exists a non-zero vector $\boldsymbol{a}_{r+1} \in \mathbb{R}^r$ such that $\boldsymbol{v}_{r+1} = \sum_{i=1}^{r} \boldsymbol{a}_{r+1,i} \boldsymbol{v}_i$.

Let $i^* = \arg\max_{i \in \mathcal{B}} |\boldsymbol{a}_{r+1,i}|$. If $|\boldsymbol{a}_{r+1,i^*}| \le 1$, then the induction hypothesis is true. Otherwise, when $|\boldsymbol{a}_{r+1,i^*}| > 1$, we have

$$\boldsymbol{v}_{i^*} = \underbrace{\frac{1}{\boldsymbol{a}_{r+1,i^*}}}_{\bar{\boldsymbol{a}}_{r+1,r+1}} \boldsymbol{v}_{r+1} - \sum_{i=1;\, i \ne i^*}^{r} \underbrace{\frac{\boldsymbol{a}_{r+1,i}}{\boldsymbol{a}_{r+1,i^*}}}_{\bar{\boldsymbol{a}}_{r+1,i}} \boldsymbol{v}_i$$

$$= \sum_{i \in \mathcal{B}^*} \bar{\boldsymbol{a}}_{r+1,i} \boldsymbol{v}_i \qquad \text{where } \mathcal{B}^* = (\mathcal{B} \cup \{r+1\}) \setminus \{i^*\},$$

i.e., we remove the item $i^*$ from $\mathcal{B}$ and include the item $r + 1$ instead. Since $|\bar{\boldsymbol{a}}_{r+1,i}| \le 1$, the induction base holds.

• *Inductive Step:* Assume the induction hypothesis is true for $m > r$, we show it is also true for $m + 1$. Without loss of generality we can assume that $\mathcal{B} = \{1, \ldots, r\}$. By induction hypothesis,

$$\boldsymbol{v}_j = \sum_{i=1}^{r} \boldsymbol{a}_{j,i} \boldsymbol{v}_i \qquad \text{with } |\boldsymbol{a}_{j,i}| \le 2^{m-r-1}, \quad \forall\, j = \{r+1, \ldots, m\}.$$

Since the case of $\boldsymbol{v}_{m+1} = 0$ trivially holds, we consider $\boldsymbol{v}_{m+1} \ne 0$. Since $\mathcal{B}$ is a basis, there exists $\boldsymbol{a}_{m+1} \ne 0$ such that $\boldsymbol{v}_{m+1} = \sum_{i=1}^{r} \boldsymbol{a}_{m+1,i} \boldsymbol{v}_i$. Let $i^* = \arg\max_{i \in \mathcal{B}} |\boldsymbol{a}_{m+1,i}|$. If $|\boldsymbol{a}_{m+1,i^*}| \le 2^{m-r}$, the induction step is done. Otherwise, for the case of $|\boldsymbol{a}_{m+1,i^*}| > 2^{m-r}$, we have

$$\boldsymbol{v}_{i^*} = \underbrace{\frac{1}{\boldsymbol{a}_{m+1,i^*}}}_{\bar{\boldsymbol{a}}_{m+1,m+1}} \boldsymbol{v}_{m+1} - \sum_{i=1;\, i \ne i^*}^{r} \underbrace{\frac{\boldsymbol{a}_{m+1,i}}{\boldsymbol{a}_{m+1,i^*}}}_{\bar{\boldsymbol{a}}_{m+1,i}} \boldsymbol{v}_i$$

$$= \sum_{i \in \mathcal{B}^*} \bar{\boldsymbol{a}}_{m+1,i} \boldsymbol{v}_i,$$

where $\mathcal{B}^* = (\mathcal{B} \cup \{m+1\}) \setminus \{i^*\}$ and clearly $|\bar{\boldsymbol{a}}_{m+1,i}| \le 1,\ \forall\, i \in \mathcal{B}^*$ according to the definition of $i^*$. For all $j \in \{r+1, \ldots, m\}$

$$\boldsymbol{v}_j = \sum_{i=1;\, i \ne i^*}^{r} \boldsymbol{a}_{j,i} \boldsymbol{v}_i + \boldsymbol{a}_{j,i^*} \boldsymbol{v}_{i^*}$$

$$= \sum_{i=1;\, i \ne i^*}^{r} \boldsymbol{a}_{j,i} \boldsymbol{v}_i + \frac{\boldsymbol{a}_{j,i^*}}{\boldsymbol{a}_{m+1,i^*}} \boldsymbol{v}_{m+1} - \sum_{i=1;\, i \ne i^*}^{r} \frac{\boldsymbol{a}_{m+1,i}\, \boldsymbol{a}_{j,i^*}}{\boldsymbol{a}_{m+1,i^*}} \boldsymbol{v}_i$$

$$= \sum_{i=1;\, i \ne i^*}^{r} \underbrace{\left( \boldsymbol{a}_{j,i} - \frac{\boldsymbol{a}_{j,i^*}\, \boldsymbol{a}_{m+1,i}}{\boldsymbol{a}_{m+1,i^*}} \right)}_{\bar{\boldsymbol{a}}_{j,i}} \boldsymbol{v}_i + \underbrace{\frac{\boldsymbol{a}_{j,i^*}}{\boldsymbol{a}_{m+1,i^*}}}_{\bar{\boldsymbol{a}}_{j,m+1}} \boldsymbol{v}_{m+1}$$

$$= \sum_{i \in \mathcal{B}^*} \bar{\boldsymbol{a}}_{j,i} \boldsymbol{v}_i.$$

It remains to show that $|\bar{\boldsymbol{a}}_{j,i}| \leq 2^{m-r}$ for all $i \in \mathcal{B}^*$, $j \in \{r+1, \ldots, m\}$. Let us first consider $i \in \mathcal{B}^* \backslash \{m+1\}$ and $j \in \{r+1, \ldots, m\}$:

$$|\bar{\boldsymbol{a}}_{j,i}| \leq |\boldsymbol{a}_{j,i}| + \left| \boldsymbol{a}_{j,i^*} \frac{\boldsymbol{a}_{m+1,i}}{\boldsymbol{a}_{m+1,i^*}} \right| \qquad \text{by triangular inequality}$$

$$\leq 2^{m-r-1} + 2^{m-r-1} \left| \frac{\boldsymbol{a}_{m+1,i}}{\boldsymbol{a}_{m+1,i^*}} \right| \qquad \text{by induction hypothesis}$$

$$\leq 2^{m-r} \qquad \text{by definition of } i^*.$$

For $i = m+1$, $|\bar{\boldsymbol{a}}_{j,m+1}| = \left| \frac{\boldsymbol{a}_{j,i^*}}{\boldsymbol{a}_{m+1,i^*}} \right| \leq \left| \frac{2^{m-r-1}}{\boldsymbol{a}_{m+1,i^*}} \right| \leq 2^{m-r}$. This concludes the inductive step and completes our proof.

$\square$

**Lemma 14.** *Let $\boldsymbol{W}_1 \in \mathbb{R}^{m \times k}$ and $\boldsymbol{W}_2 \in \mathbb{R}^{k \times n}$. Assume further that $\boldsymbol{W}_1\boldsymbol{W}_2 = \boldsymbol{U}\boldsymbol{\Sigma}\boldsymbol{V}^T$ is a singular value decomposition of the matrix product $\boldsymbol{W}_1\boldsymbol{W}_2$ with $\boldsymbol{U} \in \mathbb{R}^{m \times m}$, $\boldsymbol{V} \in \mathbb{R}^{n \times n}$, and $\boldsymbol{\Sigma} \in \mathbb{R}^{m \times n}$. Then*

$$\mathcal{M}(\cdot, \cdot) \text{ is locally open at } (\boldsymbol{W}_1, \boldsymbol{W}_2) \Leftrightarrow \mathcal{M}(\cdot, \cdot) \text{ is locally open at } (\boldsymbol{U}^T\boldsymbol{W}_1, \boldsymbol{W}_2\boldsymbol{V}).$$

*Proof.* We first show the direction " $\Rightarrow$ ". Suppose $\mathcal{M}(\cdot, \cdot)$ is locally open at $(\boldsymbol{W}_1, \boldsymbol{W}_2)$, then by definition of local openness, for any given $\epsilon > 0$, there exists $\delta > 0$ such that

$$\mathbb{B}_\delta(\boldsymbol{W}_1\boldsymbol{W}_2) \cap \mathcal{R}_\mathcal{M} \subseteq \mathcal{M}\big(\mathbb{B}_\epsilon(\boldsymbol{W}_1), \mathbb{B}_\epsilon(\boldsymbol{W}_2)\big) = \{(\boldsymbol{W}_1 + \boldsymbol{W}_1^\epsilon)(\boldsymbol{W}_2 + \boldsymbol{W}_2^\epsilon) \,|\, \|\boldsymbol{W}_1^\epsilon\| \leq \epsilon, \, \|\boldsymbol{W}_2^\epsilon\| \leq \epsilon\}. \quad (18)$$

We now show that

$$\mathbb{B}_\delta(\boldsymbol{U}^T\boldsymbol{W}_1\boldsymbol{W}_2\boldsymbol{V}) \cap \mathcal{R}_\mathcal{M} \subseteq \mathcal{M}\big(\mathbb{B}_\epsilon(\boldsymbol{U}^T\boldsymbol{W}_1), \mathbb{B}_\epsilon(\boldsymbol{W}_2\boldsymbol{V})\big).$$

Consider $\tilde{\boldsymbol{\Sigma}} \in \mathbb{B}_\delta(\boldsymbol{U}^T\boldsymbol{W}_1\boldsymbol{W}_2\boldsymbol{V}) \cap \mathcal{R}_\mathcal{M}$, i.e., $\tilde{\boldsymbol{\Sigma}} = \boldsymbol{U}^T\boldsymbol{W}_1\boldsymbol{W}_2\boldsymbol{V} + \boldsymbol{R}_\delta$ with $\text{rank}(\tilde{\boldsymbol{\Sigma}}) \leq \min\{m, k, n\}$ and $\|\boldsymbol{R}_\delta\| \leq \delta$. Since $\boldsymbol{U}$ and $\boldsymbol{V}$ are unitary matrices, we get $\boldsymbol{U}\tilde{\boldsymbol{\Sigma}}\boldsymbol{V}^T = \boldsymbol{W}_1\boldsymbol{W}_2 + \boldsymbol{U}\boldsymbol{R}_\delta\boldsymbol{V}^T$ with $\text{rank}(\boldsymbol{U}\tilde{\boldsymbol{\Sigma}}\boldsymbol{V}^T) = \text{rank}(\tilde{\boldsymbol{\Sigma}}) \leq \min\{m, k, n\}$ and $\|\boldsymbol{U}\boldsymbol{R}_\delta\boldsymbol{V}^T\| = \|\boldsymbol{R}_\delta\| \leq \delta$. According to (18), we have

$$\boldsymbol{U}\tilde{\boldsymbol{\Sigma}}\boldsymbol{V}^T \in \mathbb{B}_\delta(\boldsymbol{W}_1\boldsymbol{W}_2) \cap \mathcal{R}_\mathcal{M} \subseteq \{(\boldsymbol{W}_1 + \boldsymbol{W}_1^\epsilon)(\boldsymbol{W}_2 + \boldsymbol{W}_2^\epsilon) \,|\, \|\boldsymbol{W}_1^\epsilon\| \leq \epsilon, \, \|\boldsymbol{W}_2^\epsilon\| \leq \epsilon\}.$$

which implies,

$$\tilde{\boldsymbol{\Sigma}} \in \{(\boldsymbol{U}^T\boldsymbol{W}_1 + \boldsymbol{U}^T\boldsymbol{W}_1^\epsilon)(\boldsymbol{W}_2\boldsymbol{V} + \boldsymbol{W}_2^\epsilon\boldsymbol{V}) \,|\, \|\boldsymbol{W}_1^\epsilon\| \leq \epsilon, \, \|\boldsymbol{W}_2^\epsilon\| \leq \epsilon\}$$
$$= \{(\boldsymbol{U}^T\boldsymbol{W}_1 + \boldsymbol{U}^T\boldsymbol{W}_1^\epsilon)(\boldsymbol{W}_2\boldsymbol{V} + \boldsymbol{W}_2^\epsilon\boldsymbol{V}) \,|\, \|\boldsymbol{U}^T\boldsymbol{W}_1^\epsilon\| \leq \epsilon, \, \|\boldsymbol{W}_2^\epsilon\boldsymbol{V}\| \leq \epsilon\}.$$

Since $\tilde{\boldsymbol{\Sigma}}$ was arbitrarily chosen, we get $\mathbb{B}_\delta(\boldsymbol{U}^T\boldsymbol{W}_1\boldsymbol{W}_2\boldsymbol{V}) \cap \mathcal{R}_\mathcal{M} \subseteq \mathcal{M}\big(\mathbb{B}_\epsilon(\boldsymbol{U}^T\boldsymbol{W}_1), \mathbb{B}_\epsilon(\boldsymbol{W}_2\boldsymbol{V})\big)$.

Proving the converse direction " $\Leftarrow$ " is similar. Suppose $\mathcal{M}(\cdot, \cdot)$ is locally open at $(\boldsymbol{U}^T\boldsymbol{W}_1, \boldsymbol{W}_2\boldsymbol{V})$, then by definition of local openness, for any given $\epsilon > 0$, there exists $\delta > 0$ such that

$$\mathbb{B}_\delta(\boldsymbol{U}^T\boldsymbol{W}_1\boldsymbol{W}_2\boldsymbol{V}) \cap \mathcal{R}_\mathcal{M} \subseteq \mathcal{M}\big(\mathbb{B}_\epsilon(\boldsymbol{U}^T\boldsymbol{W}_1), \mathbb{B}_\epsilon(\boldsymbol{W}_2\boldsymbol{V})\big) \quad (19)$$
$$= \{(\boldsymbol{U}^T\boldsymbol{W}_1 + \boldsymbol{W}_1^\epsilon)(\boldsymbol{W}_2\boldsymbol{V} + \boldsymbol{W}_2^\epsilon) \,|\, \|\boldsymbol{W}_1^\epsilon\| \leq \epsilon, \, \|\boldsymbol{W}_2^\epsilon\| \leq \epsilon\}.$$

We now show that

$$\mathbb{B}_\delta(\boldsymbol{W}_1\boldsymbol{W}_2) \cap \mathcal{R}_\mathcal{M} \subseteq \mathcal{M}\big(\mathbb{B}_\epsilon(\boldsymbol{W}_1), \mathbb{B}_\epsilon(\boldsymbol{W}_2)\big).$$

Consider $\tilde{\boldsymbol{Z}} \in \mathbb{B}_\delta(\boldsymbol{W}_1\boldsymbol{W}_2) \cap \mathcal{R}_\mathcal{M}$, i.e. $\tilde{\boldsymbol{Z}} = \boldsymbol{W}_1\boldsymbol{W}_2 + \boldsymbol{R}_\delta$ with $\text{rank}(\tilde{\boldsymbol{Z}}) \leq \min\{m, k, n\}$ and $\|\boldsymbol{R}_\delta\| \leq \delta$. Since $\boldsymbol{U}$ and $\boldsymbol{V}$ are unitary matrices we get $\boldsymbol{U}^T\tilde{\boldsymbol{Z}}\boldsymbol{V} = \boldsymbol{U}^T\boldsymbol{W}_1\boldsymbol{W}_2\boldsymbol{V} + \boldsymbol{U}^T\boldsymbol{R}_\delta\boldsymbol{V}$ with $\text{rank}(\boldsymbol{U}^T\tilde{\boldsymbol{Z}}\boldsymbol{V}) = \text{rank}(\tilde{\boldsymbol{Z}}) \leq \min\{m, k, n\}$ and $\|\boldsymbol{U}^T\boldsymbol{R}_\delta\boldsymbol{V}\| = \|\boldsymbol{R}_\delta\| \leq \delta$. According to (19), we have

$$\boldsymbol{U}^T\tilde{\boldsymbol{Z}}\boldsymbol{V} \in \mathbb{B}_\delta(\boldsymbol{U}^T\boldsymbol{W}_1\boldsymbol{W}_2\boldsymbol{V}) \cap \mathcal{R}_\mathcal{M} \subseteq \mathcal{M}\big(\mathbb{B}_\epsilon(\boldsymbol{U}^T\boldsymbol{W}_1), \mathbb{B}_\epsilon(\boldsymbol{W}_2\boldsymbol{V})\big).$$

which implies,

$$\tilde{\boldsymbol{Z}} \in \{(\boldsymbol{W}_1 + \boldsymbol{U}\boldsymbol{W}_1^\epsilon)(\boldsymbol{W}_2 + \boldsymbol{W}_2^\epsilon\boldsymbol{V}^T) \,|\, \|\boldsymbol{W}_1^\epsilon\| \leq \epsilon, \, \|\boldsymbol{W}_2^\epsilon\| \leq \epsilon\}$$
$$= \{(\boldsymbol{W}_1 + \boldsymbol{U}\boldsymbol{W}_1^\epsilon)(\boldsymbol{W}_2 + \boldsymbol{W}_2^\epsilon\boldsymbol{V}^T) \,|\, \|\boldsymbol{U}\boldsymbol{W}_1^\epsilon\| \leq \epsilon, \, \|\boldsymbol{W}_2^\epsilon\boldsymbol{V}^T\| \leq \epsilon\}.$$

Since $\tilde{\boldsymbol{Z}}$ was arbitrarily chosen, we get $\mathbb{B}_\delta(\boldsymbol{W}_1\boldsymbol{W}_2) \cap \mathcal{R}_\mathcal{M} \subseteq \mathcal{M}\big(\mathbb{B}_\epsilon(\boldsymbol{W}_1), \mathbb{B}_\epsilon(\boldsymbol{W}_2)\big)$, which completes the proof.

$\square$

**Lemma 15.** *Let* $\boldsymbol{W}_1 \in \mathbb{R}^{m \times k}$ *and* $\boldsymbol{W}_2 \in \mathbb{R}^{k \times n}$. *Assume further that* $\boldsymbol{W}_1\boldsymbol{W}_2 = \boldsymbol{U}\boldsymbol{\Sigma}\boldsymbol{V}^T$ *is a singular value decomposition of the matrix product* $\boldsymbol{W}_1\boldsymbol{W}_2$ *with* $\boldsymbol{U} \in \mathbb{R}^{m \times m}$, $\boldsymbol{V} \in \mathbb{R}^{n \times n}$, *and* $\boldsymbol{\Sigma} \in \mathbb{R}^{m \times n}$. *Define* $\bar{\mathbf{W}}_1 \triangleq \mathbf{U}^T\mathbf{W}_1$ *and* $\bar{\mathbf{W}}_2 \triangleq \mathbf{W}_2\boldsymbol{V}$. *Then the condition* $(A)$ *below holds true if and only if the condition* $(B)$ *is true. Similarly, condition* $(C)$ *is true if and only if condition* $(D)$ *is true.*

$(A)$ $\exists \widehat{\mathbf{W}}_1 \in \mathbb{R}^{m \times k}$ *such that* $\widehat{\mathbf{W}}_1\mathbf{W}_2 = \mathbf{0}$ *and* $\mathbf{W}_1 + \widehat{\mathbf{W}}_1$ *is full column rank.*

$(B)$ $\exists \widetilde{\mathbf{W}}_1 \in \mathbb{R}^{m \times k}$ *such that* $\widetilde{\mathbf{W}}_1\bar{\mathbf{W}}_2 = \mathbf{0}$ *and* $\bar{\mathbf{W}}_1 + \widetilde{\mathbf{W}}_1$ *is full column rank.*

$(C)$ $\exists \widehat{\mathbf{W}}_2 \in \mathbb{R}^{k \times n}$ *such that* $\mathbf{W}_1\widehat{\mathbf{W}}_2 = \mathbf{0}$ *and* $\mathbf{W}_2 + \widehat{\mathbf{W}}_2$ *is full row rank.*

$(D)$ $\exists \widetilde{\mathbf{W}}_2 \in \mathbb{R}^{k \times n}$ *such that* $\bar{\mathbf{W}}_1\widetilde{\mathbf{W}}_2 = \mathbf{0}$ *and* $\bar{\mathbf{W}}_2 + \widetilde{\mathbf{W}}_2$ *is full row rank.*

*Proof.* Setting $\widetilde{\mathbf{W}}_1 = \mathbf{U}^T\widehat{\mathbf{W}}_1$ and $\widetilde{\mathbf{W}}_2 = \widehat{\mathbf{W}}_2\boldsymbol{V}$ leads to the desired result. $\square$

Lemma 14 and Lemma 15 imply that for proving Theorem 5, without loss of generality, we can assume that the product $\bar{\mathbf{W}}_1\bar{\mathbf{W}}_2$ is equal to a diagonal matrix. We next show in Lemma 16 that if $k < \min\{m, n\}$ and $\mathrm{rank}(\mathbf{W}_1) = \mathrm{rank}(\mathbf{W}_2)$, then statements $i, ii, iii$, and $iv$ in Theorem 5 are all equivalent.

**Lemma 16.** *Let* $\mathbf{W}_1 \in \mathbb{R}^{m \times k}$, $\mathbf{W}_2 \in \mathbb{R}^{k \times n}$ *with* $\mathrm{rank}(\mathbf{W}_1) = \mathrm{rank}(\mathbf{W}_2) = r$. *Assume further that* $k < \min\{m, n\}$. *Then, the following conditions are equivalent*

*i)* $\exists \widetilde{\mathbf{W}}_1 \in \mathbb{R}^{m \times k}$ *such that* $\widetilde{\mathbf{W}}_1\mathbf{W}_2 = \mathbf{0}$ *and* $\mathbf{W}_1 + \widetilde{\mathbf{W}}_1$ *is full column rank.*

*ii)* $\exists \widetilde{\mathbf{W}}_2 \in \mathbb{R}^{k \times n}$ *such that* $\mathbf{W}_1\widetilde{\mathbf{W}}_2 = \mathbf{0}$ *and* $\mathbf{W}_2 + \widetilde{\mathbf{W}}_2$ *is full row rank.*

*iii)* $\dim\big(\mathcal{N}(\mathbf{W}_1) \cap \mathcal{C}(\mathbf{W}_2)\big) = 0$.

*iv)* $\dim\big(\mathcal{N}(\mathbf{W}_2^T) \cap \mathcal{C}(\mathbf{W}_1^T)\big) = 0$.

*Proof.* To prove the desired result we show the equivalences $ii \Leftrightarrow iii$, and $i \Leftrightarrow iv$. Then we complete the proof by showing $iii \Leftrightarrow iv$.

We first show the direction "$ii \Rightarrow iii$". Consider $\mathbf{W}_1 \in \mathbb{R}^{m \times k}, \mathbf{W}_2 \in \mathbb{R}^{k \times n}$ with both being rank $r$ matrices. Suppose $ii$ holds, then

$$\mathcal{C}(\widetilde{\mathbf{W}}_2) \subseteq \mathcal{N}(\mathbf{W}_1) \Rightarrow \mathrm{rank}(\widetilde{\mathbf{W}}_2) \leq \dim\big(\mathcal{N}(\mathbf{W}_1)\big) = k - r. \tag{20}$$

Also,

$$k = \mathrm{rank}(\mathbf{W}_2 + \widetilde{\mathbf{W}}_2) \leq \mathrm{rank}(\mathbf{W}_2) + \mathrm{rank}(\widetilde{\mathbf{W}}_2) = r + \mathrm{rank}(\widetilde{\mathbf{W}}_2). \tag{21}$$

From inequalities (20) and (21), we get

$$k - r \leq \mathrm{rank}(\widetilde{\mathbf{W}}_2) \leq k - r \Rightarrow \mathrm{rank}(\widetilde{\mathbf{W}}_2) = k - r.$$

Note that $\dim\big(\mathcal{C}(\widetilde{\mathbf{W}}_2)\big) = \dim\big(\mathcal{N}(\mathbf{W}_1)\big)$ and $\mathcal{C}(\widetilde{\mathbf{W}}_2) \subseteq \mathcal{N}(\mathbf{W}_1)$, which implies that $\mathcal{C}(\widetilde{\mathbf{W}}_2) = \mathcal{N}(\mathbf{W}_1)$.

Then since $\mathrm{rank}(\mathbf{W}_2 + \widetilde{\mathbf{W}}_2) = \mathrm{rank}(\mathbf{W}_2) + \mathrm{rank}(\widetilde{\mathbf{W}}_2)$, we get

$$\emptyset = \mathcal{C}(\widetilde{\mathbf{W}}_2) \cap \mathcal{C}(\mathbf{W}_2) = \mathcal{N}(\mathbf{W}_1) \cap \mathcal{C}(\mathbf{W}_2) \Rightarrow \dim\big(\mathcal{N}(\mathbf{W}_1) \cap \mathcal{C}(\mathbf{W}_2)\big) = 0.$$

We now show the other direction "$ii \Leftarrow iii$". Without loss of generality, let $\mathbf{W}_2 = \left[ (\mathbf{W}_2')^{k \times r} \mathbf{A}^{r \times n-r} \;\; (\mathbf{W}_2')^{k \times r} \right]$ where columns of $\mathbf{W}_2'$ are linearly independent and let $\widetilde{\mathbf{W}}_2 = \epsilon \left[ \mathbf{w}_1^1, \ldots, \mathbf{w}_1^{k-r}, 0, \ldots, 0 \right] \in \mathbb{R}^{k \times n}$ be a rank $k-r$ matrix where $\mathbf{w}_1^i$ are unit basis of $\mathcal{N}(\mathbf{W}_1)$ which yields $\mathcal{C}(\widetilde{\mathbf{W}}_2) = \mathcal{N}(\mathbf{W}_1)$. Then since $\dim\big(\mathcal{N}(\mathbf{W}_1) \cap \mathcal{C}(\mathbf{W}_2)\big) = 0$, we get $\operatorname{rank}(\mathbf{W}_2 + \widetilde{\mathbf{W}}_2) = k$ for generic choice of $\epsilon$. This completes the proof.

Note that by setting $\mathbf{W}_1 = \mathbf{W}_2^T$ and $\mathbf{W}_2 = \mathbf{W}_1^T$, the same proof can be used to show $i \Leftrightarrow iv$. Next, we will prove the equivalence $iii \Leftrightarrow iv$. Notice that

$$\dim\Big(\operatorname{span}\big(\mathcal{N}(\mathbf{W}_1) \cup \mathcal{C}(\mathbf{W}_2)\big)\Big)$$
$$= \dim\big(\mathcal{N}(\mathbf{W}_1)\big) + \dim\big(\mathcal{C}(\mathbf{W}_2)\big) - \dim\big(\mathcal{N}(\mathbf{W}_1) \cap \mathcal{C}(\mathbf{W}_2)\big)$$
$$= k - r + r - \dim\big(\mathcal{N}(\mathbf{W}_1) \cap \mathcal{C}(\mathbf{W}_2)\big)$$
$$= k - \dim\big(\mathcal{N}(\mathbf{W}_1) \cap \mathcal{C}(\mathbf{W}_2)\big).$$

Thus,

$$\dim\big(\mathcal{N}(\mathbf{W}_1) \cap \mathcal{C}(\mathbf{W}_2)\big) \neq 0 \Leftrightarrow \dim\Big(\operatorname{span}\big(\mathcal{N}(\mathbf{W}_1) \cup \mathcal{C}(\mathbf{W}_2)\big)\Big) < k$$
$$\Leftrightarrow \exists\, \mathbf{a} \neq 0 \text{ such that } \mathbf{a} \perp \mathcal{C}(\mathbf{W}_2), \text{ and } \mathbf{a} \perp \mathcal{N}(\mathbf{W}_1)$$
$$\Leftrightarrow \exists\, \mathbf{a} \neq 0 \text{ such that } \mathbf{a} \in \mathcal{N}(\mathbf{W}_2^T), \text{ and } \mathbf{a} \in \mathcal{C}(\mathbf{W}_1^T)$$
$$\Leftrightarrow \dim\big(\mathcal{N}(\mathbf{W}_2^T) \cap \mathcal{C}(\mathbf{W}_1^T)\big) \neq 0,$$

which completes the proof. $\qquad\square$

The next result shows that if the conditions in Lemma 16 hold, then $\operatorname{rank}(\mathbf{W}_1) = \operatorname{rank}(\mathbf{W}_2)$. Moreover, for $r \triangleq \operatorname{rank}(\mathbf{W}_1 \mathbf{W}_2)$, the last $n - r$ rows of $\mathbf{U}^T \mathbf{W}_1$ and last $n - r$ columns of $\mathbf{W}_2 \mathbf{V}$ are all zeros, where the columns of $\mathbf{U} \in \mathbb{R}^{m \times m}$ and the columns of $\mathbf{V} \in \mathbb{R}^{n \times n}$ are respectively the left and right singular vectors of the product $\mathbf{W}_1 \mathbf{W}_2$.

**Lemma 17.** *Let $\mathbf{W}_1 \in \mathbb{R}^{m \times k}$, $\mathbf{W}_2 \in \mathbb{R}^{k \times n}$ with $k < \min\{m, n\}$ and let $r \triangleq \operatorname{rank}(\mathbf{W}_1 \mathbf{W}_2)$. Assume further that $\mathbf{W}_1 \mathbf{W}_2 = \mathbf{U} \mathbf{\Sigma} \mathbf{V}^T$ is an SVD decomposition of $\mathbf{W}_1 \mathbf{W}_2$ with $\mathbf{U} \in \mathbb{R}^{m \times m}$, and $\mathbf{V} \in \mathbb{R}^{n \times n}$, and $\mathbf{\Sigma} \in \mathbb{R}^{m \times n}$. If*

$$\begin{cases} i)\; \exists\, \widetilde{\mathbf{W}}_1 \in \mathbb{R}^{m \times k} \text{ such that } \widetilde{\mathbf{W}}_1 \mathbf{W}_2 = \mathbf{0} \text{ and } \mathbf{W}_1 + \widetilde{\mathbf{W}}_1 \text{ is full column rank.} \\ \qquad\qquad\qquad\qquad\qquad\qquad\text{and} \\ ii)\; \exists\, \widetilde{\mathbf{W}}_2 \in \mathbb{R}^{k \times n} \text{ such that } \mathbf{W}_1 \widetilde{\mathbf{W}}_2 = \mathbf{0} \text{ and } \mathbf{W}_2 + \widetilde{\mathbf{W}}_2 \text{ is full row rank.} \end{cases}$$

*then*

$$\operatorname{rank}(\mathbf{W}_1) = \operatorname{rank}(\mathbf{W}_2), \quad \big(\mathbf{W}_2 \mathbf{V}\big)_{:,r+1:n} = 0, \quad and \quad \big(\mathbf{U}^T \mathbf{W}_1\big)_{r+1:n,:} = 0.$$

*Proof.* Suppose that $ii)$ holds, then

$$\mathcal{C}(\widetilde{\mathbf{W}}_2) \subseteq \mathcal{N}(\mathbf{W}_1) \Rightarrow \operatorname{rank}(\widetilde{\mathbf{W}}_2) \leq \dim\big(\mathcal{N}(\mathbf{W}_1)\big) = k - \operatorname{rank}(\mathbf{W}_1). \tag{22}$$

Also,

$$k = \operatorname{rank}(\mathbf{W}_2 + \widetilde{\mathbf{W}}_2) \leq \operatorname{rank}(\mathbf{W}_2) + \operatorname{rank}(\widetilde{\mathbf{W}}_2). \tag{23}$$

From inequalities (22) and (23), we get

$$k - \operatorname{rank}(\mathbf{W}_2) \leq \operatorname{rank}(\widetilde{\mathbf{W}}_2) \leq k - \operatorname{rank}(\mathbf{W}_1) \Rightarrow \operatorname{rank}(\mathbf{W}_2) \geq \operatorname{rank}(\mathbf{W}_1). \tag{24}$$

Similarly, condition $i)$ implies

$$\mathcal{C}(\widetilde{\mathbf{W}}_1^T) \subseteq \mathcal{N}(\bar{\mathbf{W}}_2^T) \Rightarrow \operatorname{rank}(\widetilde{\mathbf{W}}_1) \leq \dim\big(\mathcal{N}(\mathbf{W}_2^T)\big) = k - \operatorname{rank}(\mathbf{W}_2). \tag{25}$$

Also,

$$k = \operatorname{rank}(\boldsymbol{W}_1 + \widetilde{\mathbf{W}}_1) \le \operatorname{rank}(\boldsymbol{W}_1) + \operatorname{rank}(\widetilde{\mathbf{W}}_1). \tag{26}$$

From inequalities (25) and (26), we get

$$k - \operatorname{rank}(\boldsymbol{W}_1) \le \operatorname{rank}(\widetilde{\mathbf{W}}_1) \le k - \operatorname{rank}(\boldsymbol{W}_2) \Rightarrow \operatorname{rank}(\boldsymbol{W}_1) \ge \operatorname{rank}(\boldsymbol{W}_2). \tag{27}$$

From inequalities (24) and (27), we get $\operatorname{rank}(\boldsymbol{W}_1) = \operatorname{rank}(\boldsymbol{W}_2)$, which combined with Lemma 16 implies $\dim\big(\mathcal{N}(\boldsymbol{W}_1) \cap \mathcal{C}(\boldsymbol{W}_2)\big) = 0$. Let $r \triangleq \operatorname{rank}(\mathbf{W}_1\mathbf{W}_2)$. It directly follows from the SVD decomposition of the matrix $\boldsymbol{W}_1\boldsymbol{W}_2$, that $\boldsymbol{U}^T\boldsymbol{W}_1\big(\boldsymbol{W}_2\boldsymbol{V}\big)_{:,r+1:n} = \boldsymbol{\Sigma}_{:,r+1:n} = \mathbf{0}$, or equivalently $\boldsymbol{W}_1\big(\boldsymbol{W}_2\boldsymbol{V}\big)_{:,r+1:n} = \mathbf{0}$. On the other hand, since $\mathcal{C}\big(\boldsymbol{W}_2\boldsymbol{V}_{:,r+1:n}\big) \subset \mathcal{C}\big(\boldsymbol{W}_2\big)$ and $\mathcal{N}\big(\boldsymbol{W}_1\big) \cap \mathcal{C}\big(\boldsymbol{W}_2\big) = \emptyset$, we conclude that $\big(\boldsymbol{W}_2\boldsymbol{V}\big)_{:,r+1:n} = \mathbf{0}$. Similarly, one can show that $\big(\boldsymbol{U}^T\boldsymbol{W}_1\big)_{r+1:n,:} = \mathbf{0}$. $\qquad\square$

**Proposition 18.** *Let $\mathcal{M}(\mathbf{W}_1, \mathbf{W}_2) = \mathbf{W}_1\mathbf{W}_2$ be the matrix product mapping with $\mathbf{W}_1 \in \mathbb{R}^{m \times k}$, $\mathbf{W}_2 \in \mathbb{R}^{k \times n}$, and $k < \min\{m, n\}$. Then, $\mathcal{M}(\cdot, \cdot)$ is locally open in its range $\mathcal{R}_\mathcal{M} \triangleq \{\mathbf{Z} \in \mathbb{R}^{m \times n} : \operatorname{rank}(\mathbf{Z}) \le k\}$ at the point $(\bar{\mathbf{W}}_1, \bar{\mathbf{W}}_2)$ if and only if the following two conditions are satisfied:*

$$\begin{cases} i)\ \exists\, \widetilde{\mathbf{W}}_1 \in \mathbb{R}^{m \times k} \text{ such that } \widetilde{\mathbf{W}}_1\bar{\mathbf{W}}_2 = \mathbf{0} \text{ and } \bar{\mathbf{W}}_1 + \widetilde{\mathbf{W}}_1 \text{ is full column rank.} \\ \qquad\qquad\qquad\qquad\qquad and \\ ii)\ \exists\, \widetilde{\mathbf{W}}_2 \text{ such that } \bar{\mathbf{W}}_1\widetilde{\mathbf{W}}_2 = \mathbf{0} \text{ and } \bar{\mathbf{W}}_2 + \widetilde{\mathbf{W}}_2 \text{ is full row rank.} \end{cases}$$

*Proof.* First of all, according to Lemma 14 and Lemma 15, without loss of generality we can assume that the matrix product $\bar{\mathbf{W}}_1\bar{\mathbf{W}}_2$ is of diagonal form.

Let us start by first proving the "only if" direction. Notice that the result clearly holds when $\operatorname{rank}(\bar{\mathbf{W}}_1) = \operatorname{rank}(\bar{\mathbf{W}}_2) = k$ by choosing $\widetilde{\mathbf{W}}_1 = \widetilde{\mathbf{W}}_2 = \mathbf{0}$. Moreover, the mapping $\mathcal{M}(\cdot, \cdot)$ cannot be locally open if only one of the matrices $\bar{\mathbf{W}}_1$ or $\bar{\mathbf{W}}_2$ is rank deficient. To see this, let us assume that $\bar{\mathbf{W}}_1$ is full column rank, while $\bar{\mathbf{W}}_2$ is rank deficient. Assume further that the mapping $\mathcal{M}(\cdot, \cdot)$ is locally open at $(\bar{\mathbf{W}}_1, \bar{\mathbf{W}}_2)$, it follows from the definition of openness that the mapping $\mathcal{M}^1(\mathbf{W}_1, \mathbf{W}_2^1) \triangleq \mathbf{W}_1\mathbf{W}_2^1$ is locally open at $(\bar{\mathbf{W}}_1, \bar{\mathbf{W}}_2^1)$ where $\bar{\mathbf{W}}_2^1 \triangleq (\bar{\mathbf{W}}_2)_{:,1:k}$ only contains the first $k$ columns of $\bar{\mathbf{W}}_2$. Since the range of the mapping $\mathcal{M}^1$ at $(\bar{\mathbf{W}}_1, \bar{\mathbf{W}}_2^1)$ is the entire space $\mathbb{R}^{m \times k}$, Proposition 4 implies that

$$\begin{cases} \exists\, \widetilde{\mathbf{W}}_1 \text{ such that } \widetilde{\mathbf{W}}_1\bar{\mathbf{W}}_2^1 = \mathbf{0} \text{ and } \bar{\mathbf{W}}_1 + \widetilde{\mathbf{W}}_1 \text{ is full row rank.} \\ \qquad\qquad\qquad\qquad\qquad or \\ \exists\, \widetilde{\mathbf{W}}_2^1 \text{ such that } \bar{\mathbf{W}}_1\widetilde{\mathbf{W}}_2^1 = \mathbf{0} \text{ and } \bar{\mathbf{W}}_2^1 + \widetilde{\mathbf{W}}_2^1 \text{ is full rank.} \end{cases}$$

Moreover, since $\bar{\mathbf{W}}_1 \in \mathbb{R}^{m \times k}$ and $m > k$, it is impossible for $\widetilde{\mathbf{W}}_1 + \bar{\mathbf{W}}_1$ to be full row rank. On the other hand, since $\bar{\mathbf{W}}_1$ is full column rank, $\bar{\mathbf{W}}_1\widetilde{\mathbf{W}}_2^1 = 0$ implies that $\widetilde{\mathbf{W}}_2^1 = 0$; and hence $\bar{\mathbf{W}}_2^1 + \widetilde{\mathbf{W}}_2^1$ is not full column rank. Hence none of the above two conditions can hold and consequently, $\mathcal{M}(\cdot, \cdot)$ cannot be open at the point $(\bar{\mathbf{W}}_1, \mathbf{W}_2^1)$ in this case. Similarly, we can show that when $\bar{\mathbf{W}}_1$ is rank deficient and $\bar{\mathbf{W}}_2$ is full row rank, the mapping $\mathcal{M}(\cdot, \cdot)$ cannot be locally open. Hence, if $\bar{\mathbf{W}}_1$ and $\bar{\mathbf{W}}_2$ are not both full rank, then they both should be rank deficient.

Assume that the matrices $\bar{\mathbf{W}}_1$ and $\bar{\mathbf{W}}_2$ are both rank deficient and $\mathcal{M}(\cdot, \cdot)$ is locally open at $(\bar{\mathbf{W}}_1, \bar{\mathbf{W}}_2)$. It follows that $\mathcal{M}^1(\mathbf{W}_1, \mathbf{W}_2^1) \triangleq \mathbf{W}_1\mathbf{W}_2^1$ is locally open at $(\bar{\mathbf{W}}_1, \bar{\mathbf{W}}_2^1)$. By Proposition 4, and since there does not exist $\widetilde{\mathbf{W}}_1$ such that $\bar{\mathbf{W}}_1 + \widetilde{\mathbf{W}}_1$ is full row rank, there should exist $\widetilde{\mathbf{W}}_2^1$ such that $\bar{\mathbf{W}}_1\widetilde{\mathbf{W}}_2^1 = \mathbf{0}$ and $\bar{\mathbf{W}}_2^1 + \widetilde{\mathbf{W}}_2^1$ is full rank. Defining $\widetilde{\mathbf{W}}_2 \triangleq \begin{bmatrix} \widetilde{\mathbf{W}}_2^1 & | & \mathbf{0} \end{bmatrix}$, we satisfy the desired condition $ii)$.

Similarly, by looking at the transpose of the mapping $\mathcal{M}$, we can show that condition $i)$ is true when $\mathcal{M}$ is locally open.

We now prove the "if" direction. Suppose $i)$ and $ii)$ hold, i.e.,

$$\begin{cases} \exists\, \widetilde{\mathbf{W}}_1 \text{ such that } \widetilde{\mathbf{W}}_1\bar{\mathbf{W}}_2 = \mathbf{0} \text{ and } \bar{\mathbf{W}}_1 + \widetilde{\mathbf{W}}_1 \text{ is full column rank.} \\ \qquad\qquad\qquad\qquad\qquad and \\ \exists\, \widetilde{\mathbf{W}}_2 \text{ such that } \bar{\mathbf{W}}_1\widetilde{\mathbf{W}}_2 = \mathbf{0} \text{ and } \bar{\mathbf{W}}_2 + \widetilde{\mathbf{W}}_2 \text{ is full row rank.} \end{cases}$$

Let $\boldsymbol{\Sigma} = \bar{\boldsymbol{W}}_1 \bar{\boldsymbol{W}}_2 = [\ \boldsymbol{\Sigma}_{:,1:r}\quad 0\ ]$ be a rank $r$ matrix. Lemma 17 implies that $\mathrm{rank}(\bar{\mathbf{W}}_1) = \mathrm{rank}(\bar{\mathbf{W}}_2)$, and the last $n-r$ columns of $\bar{\boldsymbol{W}}_2$ are all zeros. We need to show that for any given $\epsilon > 0$, there exists $\delta > 0$, such that

$$\mathbb{B}_\delta(\bar{\boldsymbol{W}}_1\bar{\boldsymbol{W}}_2) \cap \mathcal{R}_\mathcal{M} \subseteq \mathcal{M}\Big(\mathbb{B}_\epsilon(\bar{\boldsymbol{W}}_1), \mathbb{B}_\epsilon(\bar{\boldsymbol{W}}_2)\Big).$$

Consider a perturbed matrix $\tilde{\boldsymbol{\Sigma}} \in \mathbb{B}_\delta(\boldsymbol{\Sigma}) \cap \mathcal{R}_\mathcal{M}$, we show that $\tilde{\boldsymbol{\Sigma}} \in \mathcal{M}\Big(\mathbb{B}_\epsilon(\bar{\boldsymbol{W}}_1), \mathbb{B}_\epsilon(\bar{\boldsymbol{W}}_2)\Big)$. Without loss of generality, and by permuting the columns of $\tilde{\boldsymbol{\Sigma}}$ if necessary, $\tilde{\boldsymbol{\Sigma}}$ can be expressed as

$$\tilde{\boldsymbol{\Sigma}} = \left[\ \underbrace{\boldsymbol{\Sigma}_{:,1:r} + \boldsymbol{R}_\delta^1}_{m \times r}\ \Bigg|\ \underbrace{\boldsymbol{R}_\delta^2}_{m \times (k-r)}\ \Bigg|\ \underbrace{(\boldsymbol{\Sigma}_{:,1:r} + \boldsymbol{R}_\delta^1)\boldsymbol{A}_1 + \boldsymbol{R}_\delta^2 \boldsymbol{A}_2}_{m \times (n-k)}\ \right].$$

Here $\boldsymbol{A}_1 \in \mathbb{R}^{r \times (n-k)}$ and $\boldsymbol{A}_2 \in \mathbb{R}^{(k-r) \times (n-k)}$ exist since $\mathrm{rank}(\tilde{\boldsymbol{\Sigma}}) \le k$. Moreover, $\|\tilde{\boldsymbol{\Sigma}} - \boldsymbol{\Sigma}\| \le \delta$ implies that the perturbed matrix

$$\boldsymbol{R}_\delta \triangleq \left[\ \boldsymbol{R}_\delta^1\ \big|\ \boldsymbol{R}_\delta^2\ \big|\ (\boldsymbol{\Sigma}_{:,1:r} + \boldsymbol{R}_\delta^1)\boldsymbol{A}_1 + \boldsymbol{R}_\delta^2 \boldsymbol{A}_2\ \right]$$

has norm less than or equal $\delta$, i.e. $\|\boldsymbol{R}_\delta\| \le \delta$.

Since $\mathrm{rank}(\bar{\mathbf{W}}_2 + \widetilde{\mathbf{W}}_2) = k$, there exist a unitary basis set $\{\tilde{\mathbf{w}}_2^1, \ldots, \tilde{\mathbf{w}}_2^{k-r}\}$ for $\widetilde{\mathbf{W}}_2$ such that $span\{\tilde{\mathbf{w}}_2^1, \ldots, \tilde{\mathbf{w}}_2^{k-r}\} \cap \mathcal{C}(\bar{\mathbf{W}}_2) = 0$. Define

$$\widetilde{\mathbf{W}}_2^1 \triangleq \frac{\epsilon}{n2^{n+1}} \left[\ \overbrace{0}^{k \times r}\quad \overbrace{\tilde{\mathbf{w}}_2^1 \ldots, \tilde{\mathbf{w}}_2^{k-r}}^{k \times (k-r)}\ \right], \tag{28}$$

and let us form the matrix $\bar{\mathbf{W}}_2^1 \in \mathbb{R}^{k \times k}$ using the first $k$ columns of $\bar{\mathbf{W}}_2$. Since the last $n-r$ columns of the matrix $\bar{\mathbf{W}}_2$ are zero, $\widetilde{\mathbf{W}}_2^1 + \bar{\mathbf{W}}_2^1$ is a full rank $k \times k$ matrix and $\bar{\mathbf{W}}_1 \widetilde{\mathbf{W}}_2^1 = 0$. Let us define

$$\bar{\mathbf{W}}_1^0 \triangleq \left[\ \boldsymbol{R}_\delta^1\ \big|\ \boldsymbol{R}_\delta^2\ \right](\widetilde{\mathbf{W}}_2^1 + \bar{\mathbf{W}}_2^1)^{-1},$$

and

$$\bar{\mathbf{W}}_2^0 \triangleq \left[\ \widetilde{\mathbf{W}}_2^1\ \Big|\ \big(\bar{\mathbf{W}}_2^1 + \widetilde{\mathbf{W}}_2^1\big)_{:,1:r}\boldsymbol{A}_1 + \big(\bar{\mathbf{W}}_2^1 + \widetilde{\mathbf{W}}_2^1\big)_{:,r+1:k}\boldsymbol{A}_2\ \right].$$

Using this definition, we have

$$(\bar{\boldsymbol{W}}_1 + \bar{\mathbf{W}}_1^0)(\bar{\boldsymbol{W}}_2 + \bar{\mathbf{W}}_2^0)$$
$$= \left[(\bar{\boldsymbol{W}}_1 + \bar{\mathbf{W}}_1^0)(\bar{\boldsymbol{W}}_2 + \bar{\mathbf{W}}_2^0)_{:,1:k}\ \Big|\ (\bar{\boldsymbol{W}}_1 + \bar{\mathbf{W}}_1^0)(\bar{\boldsymbol{W}}_2 + \bar{\mathbf{W}}_2^0)_{:,k+1:n}\right]$$
$$= \left[(\bar{\boldsymbol{W}}_1 + \bar{\mathbf{W}}_1^0)(\bar{\mathbf{W}}_2^1 + \widetilde{\mathbf{W}}_2^1)\ \Big|\ (\bar{\boldsymbol{W}}_1 + \bar{\mathbf{W}}_1^0)(\bar{\boldsymbol{W}}_2 + \bar{\mathbf{W}}_2^0)_{:,k+1:n}\right]$$
$$= \left[\ \bar{\boldsymbol{\Sigma}}_{:,1:k} + \underbrace{\bar{\boldsymbol{W}}_1 \widetilde{\mathbf{W}}_2^1}_{=0} + \left[\ \boldsymbol{R}_\delta^1\ \big|\ \boldsymbol{R}_\delta^2\ \right](\bar{\mathbf{W}}_2^1 + \widetilde{\mathbf{W}}_2^1)^{-1}(\bar{\mathbf{W}}_2^1 + \widetilde{\mathbf{W}}_2^1)\ \Bigg|\ \underbrace{\mathbf{0}}_{m \times (n-k)}\ \right]$$
$$+ \left[\ \underbrace{\mathbf{0}}_{m \times k}\ \Bigg|\ (\bar{\boldsymbol{W}}_1 + \bar{\mathbf{W}}_1^0)\left[\ \left[\ \big(\bar{\mathbf{W}}_2^1 + \widetilde{\mathbf{W}}_2^1\big)_{:,1:r}\quad\quad \big(\bar{\mathbf{W}}_2^1 + \widetilde{\mathbf{W}}_2^1\big)_{:,r+1:k}\ \right]\left[\begin{array}{c}\boldsymbol{A}_1 \\ \boldsymbol{A}_2\end{array}\right]\right]\ \right]$$
$$= \bar{\boldsymbol{W}}_1 \bar{\boldsymbol{W}}_2 + \left[\ \boldsymbol{R}_\delta^1\ \big|\ \boldsymbol{R}_\delta^2\ \big|\ (\boldsymbol{\Sigma}_{:,1:r} + \boldsymbol{R}_\delta^1)\boldsymbol{A}_1 + \boldsymbol{R}_\delta^2 \boldsymbol{A}_2\ \right]$$
$$= \bar{\boldsymbol{W}}_1 \bar{\boldsymbol{W}}_2 + \boldsymbol{R}_\delta$$
$$= \tilde{\boldsymbol{\Sigma}}. \tag{29}$$

$\square$

To complete the proof, it remains to show that for any $\epsilon > 0$, we can choose $\delta$ small enough such that $\|\bar{\mathbf{W}}_1^0\| \le \epsilon$ and $\|\bar{\mathbf{W}}_2^0\| \le \epsilon$. In other words, we will show $\tilde{\boldsymbol{\Sigma}} \in \mathcal{M}\Big(\mathbb{B}_\epsilon(\bar{\boldsymbol{W}}_1), \mathbb{B}_\epsilon(\bar{\boldsymbol{W}}_2)\Big)$.

Let $\tilde{r}$, with $k \geq \tilde{r} \geq r$, be the rank of $\tilde{\boldsymbol{\Sigma}}$. According to Lemma 13 and by possibly permuting the columns, $\tilde{\boldsymbol{\Sigma}}$ can be expressed as

$$\tilde{\boldsymbol{\Sigma}} = \left[ \begin{array}{c|c} \tilde{\boldsymbol{\Sigma}}_1 & \tilde{\boldsymbol{\Sigma}}_1 \bar{\boldsymbol{A}} \end{array} \right],$$

where $\tilde{\boldsymbol{\Sigma}}_1 \in \mathbb{R}^{m \times \tilde{r}}$ is full column rank, and $\bar{\boldsymbol{A}}$ has a bounded norm $\|\bar{\boldsymbol{A}}\| \leq n2^{n-\tilde{r}-1}$. Notice that for given $\bar{\mathbf{W}}_1^0$ and $\bar{\mathbf{W}}_2^0$ satisfying (29), permuting the columns of $\tilde{\boldsymbol{\Sigma}}$ corresponds to permuting the columns of $(\bar{\boldsymbol{W}}_2 + \bar{\boldsymbol{W}}_2^0)$. If we can show that the first $r$ columns are not among the permuted ones, then using the fact that $\bar{\boldsymbol{W}}_2$ has only its first $r$ columns non-zero, it follows that the permutation of the columns of $\tilde{\boldsymbol{\Sigma}}$ corresponds to the same permutation of the columns of $\bar{\mathbf{W}}_2^0$. Moreover, if the first $r$ columns are not among the permuted ones, then without loss of generality we can express the perturbed matrix

$$\tilde{\boldsymbol{\Sigma}} = \left[ \begin{array}{c|c|c} \underbrace{\boldsymbol{\Sigma}_{:,1:r} + \boldsymbol{R}_\delta^1}_{m \times r} & \underbrace{\boldsymbol{R}_\delta^2}_{m \times (k-r)} & \underbrace{(\boldsymbol{\Sigma}_{:,1:r} + \boldsymbol{R}_\delta^1)\bar{\boldsymbol{A}}_1 + \boldsymbol{R}_\delta^2 \bar{\boldsymbol{A}}_2}_{m \times (n-k)} \end{array} \right],$$

and the perturbation matrix

$$\boldsymbol{R}_\delta = \left[ \begin{array}{c|c|c} \underbrace{\boldsymbol{R}_\delta^1}_{m \times r} & \underbrace{\boldsymbol{R}_\delta^2}_{m \times (k-r)} & \underbrace{(\boldsymbol{\Sigma}_{:,1:r} + \boldsymbol{R}_\delta^1)\bar{\boldsymbol{A}}_1 + \boldsymbol{R}_\delta^2 \bar{\boldsymbol{A}}_2}_{m \times (n-k)} \end{array} \right],$$

where $\left[ \begin{array}{c} \bar{\boldsymbol{A}}_1 \\ \bar{\boldsymbol{A}}_2 \end{array} \right] = \bar{\boldsymbol{A}}$ has a bounded norm.

We now show that the first $r$ columns of $\tilde{\boldsymbol{\Sigma}}$ before permutation $\boldsymbol{\Sigma}_{:,1:r} + \boldsymbol{R}_\delta^1 \subseteq \tilde{\boldsymbol{\Sigma}}_1$. Assume the contrary, then there exists at least a column $\boldsymbol{\Sigma}_{:,j} + \left( \boldsymbol{R}_\delta^1 \right)_{:,j}$ that is not a column of $\tilde{\boldsymbol{\Sigma}}_1$, which implies $\boldsymbol{\Sigma}_{:,j} + \left( \boldsymbol{R}_\delta^1 \right)_{:,j}$ is a column of $\tilde{\boldsymbol{\Sigma}}_1 \bar{\boldsymbol{A}}$. Without loss of generality let $\boldsymbol{\Sigma}_{:,j} + \left( \boldsymbol{R}_\delta^1 \right)_{:,j} = \tilde{\boldsymbol{\Sigma}}_1 \bar{\boldsymbol{A}}_{:,1}$. It follows that

$$\boldsymbol{\Sigma}_{j,j} + \left( \boldsymbol{R}_\delta^1 \right)_{j,j} = (\tilde{\boldsymbol{\Sigma}}_1)_{j,:} \bar{\boldsymbol{A}}_{:,1}.$$

But since $\boldsymbol{\Sigma}_{j,j} + \left( \boldsymbol{R}_\delta^1 \right)_{j,j}$ is a non-zero perturbed singular value, and since elements of $(\tilde{\boldsymbol{\Sigma}}_1)_{j,:}$ are all of order $\delta$, then by choosing $\delta$ sufficiently small, we get $\|\bar{\boldsymbol{A}}\| > 2^{n-\tilde{r}-1}$, which contradicts the bound we have on $\bar{\boldsymbol{A}}$.

We now obtain an upper-bound on $\|\bar{\mathbf{W}}_2^0\|$. Since the norm of $\bar{\boldsymbol{A}}$ is bounded, the norm of $\bar{\boldsymbol{A}}_2$ is also bounded by some constant $K \triangleq n2^n > n2^{n-\tilde{r}-1}$. Hence,

$$\begin{aligned} \delta \geq \|\boldsymbol{R}_\delta\| &\geq \|(\boldsymbol{\Sigma}_{:,1:r} + \boldsymbol{R}_\delta^1)\bar{\boldsymbol{A}}_1 + \boldsymbol{R}_\delta^2 \bar{\boldsymbol{A}}_2\| \\ &\geq \|(\boldsymbol{\Sigma}_{:,1:r} + \boldsymbol{R}_\delta^1)\bar{\boldsymbol{A}}_1\| - \|\boldsymbol{R}_\delta^2 \bar{\boldsymbol{A}}_2\| \\ &\geq \|(\boldsymbol{\Sigma}_{:,1:r} + \boldsymbol{R}_\delta^1)\bar{\boldsymbol{A}}_1\| - K\delta \\ &\geq \frac{\sigma_{\min}}{2}\|\bar{\boldsymbol{A}}_1\| - K\delta, \end{aligned}$$

where $\sigma_{\min}$ is the minimum singular value of the full column rank matrix $\boldsymbol{\Sigma}_{:,1:r}$ which is bounded away from zero. Here, we have chosen $\delta < \sigma_{\min}/2$ so that $\|(\boldsymbol{\Sigma}_{:,1:r} + \boldsymbol{R}_\delta^1)\bar{\boldsymbol{A}}_1\| \leq \frac{\sigma_{\min}}{2}\|\bar{\boldsymbol{A}}_1\|$. Rearranging the terms, we obtain

$$\|\bar{\boldsymbol{A}}_1\| \leq \frac{2(1+K)\delta}{\sigma_{\min}}.$$

Thus, for some constant $C \triangleq \|\bar{\mathbf{W}}_2^1\|$, we obtain

$$\|\bar{\mathbf{W}}_2^0\|^2 \le \|\widetilde{\mathbf{W}}_2^1\|^2 + \|\bar{\mathbf{W}}_2^1\|^2 \|\bar{\boldsymbol{A}}_1\|^2 + \|\widetilde{\mathbf{W}}_2^1\|^2 \|\bar{\boldsymbol{A}}_2\|^2 \qquad \text{(by triangular inequality and Cauchy Shwarz)}$$

$$\le \frac{\epsilon^2}{4n^2 2^{2n}} + \frac{\epsilon^2 K^2}{4n^2 2^{2n}} + \delta^2 C^2 \left(\frac{2+2K}{\sigma_{min}}\right)^2$$

$$\le \frac{\epsilon^2}{4K^2} + \frac{\epsilon^2 K^2}{4K^2} + \delta^2 C^2 \left(\frac{2+2K}{\sigma_{min}}\right)^2$$

$$\le \epsilon^2/2 + \delta^2 C^2 \left(\frac{2+2K}{\sigma_{min}}\right)^2.$$

For a given $\epsilon > 0$, choose

$$\delta \le \min\left\{ \frac{\epsilon}{1 + \max\left\{ \|(\bar{\mathbf{W}}_2^1 + \widetilde{\mathbf{W}}_2^1)^{-1}\|, \sqrt{2}\,C\left(\frac{2+2K}{\sigma_{min}}\right) \right\}}, \sigma_{\min}/2 \right\}.$$

This choice of $\delta$ leads to $\|\bar{\mathbf{W}}_2^0\| \le \epsilon$. Moreover,

$$\|\bar{\mathbf{W}}_1^0\| \le \|\boldsymbol{R}_\delta\| \, \|(\mathbf{W}_2^1 + \widetilde{\mathbf{W}}_2^1)^{-1}\|$$

$$\le \delta \|(\mathbf{W}_2^1 + \widetilde{\mathbf{W}}_2^1)^{-1}\|$$

$$\le \frac{\epsilon \|(\mathbf{W}_2^1 + \widetilde{\mathbf{W}}_2^1)^{-1}\|}{1 + \|(\mathbf{W}_2^1 + \widetilde{\mathbf{W}}_2^1)^{-1}\|}$$

$$\le \epsilon,$$

which completes the proof.

We now use Proposition 18, Lemma 16, and Lemma 17 to complete the proof of Theorem 5 restated below.

**Theorem 5 [Restated]**: Let $\mathcal{M}(\mathbf{W}_1, \mathbf{W}_2) = \mathbf{W}_1 \mathbf{W}_2$ denote the matrix multiplication mapping with $\mathbf{W}_1 \in \mathbb{R}^{m \times k}$ and $\mathbf{W}_2 \in \mathbb{R}^{k \times n}$. Assume $k < \min\{m, n\}$. Then if $\mathrm{rank}(\bar{\mathbf{W}}_1) \ne \mathrm{rank}(\bar{\mathbf{W}}_2)$, $\mathcal{M}(\cdot, \cdot)$ is not locally open at $(\bar{\mathbf{W}}_1, \bar{\mathbf{W}}_2)$. Else, if $\mathrm{rank}(\bar{\mathbf{W}}_1) = \mathrm{rank}(\bar{\mathbf{W}}_2)$, then the following statements are equivalent:

i) $\exists\, \widetilde{\mathbf{W}}_1 \in \mathbb{R}^{m \times k}$ such that $\widetilde{\mathbf{W}}_1 \bar{\mathbf{W}}_2 = \mathbf{0}$ and $\bar{\mathbf{W}}_1 + \widetilde{\mathbf{W}}_1$ is full column rank.

ii) $\exists\, \widetilde{\mathbf{W}}_2 \in \mathbb{R}^{k \times n}$ such that $\bar{\mathbf{W}}_1 \widetilde{\mathbf{W}}_2 = \mathbf{0}$ and $\bar{\mathbf{W}}_2 + \widetilde{\mathbf{W}}_2$ is full row rank.

iii) $\dim\left(\mathcal{N}(\bar{\mathbf{W}}_1) \cap \mathcal{C}(\bar{\mathbf{W}}_2)\right) = 0.$

iv) $\dim\left(\mathcal{N}(\bar{\mathbf{W}}_2^T) \cap \mathcal{C}(\bar{\mathbf{W}}_1^T)\right) = 0.$

v) $\mathcal{M}(\cdot, \cdot)$ is locally open at $(\bar{\mathbf{W}}_1, \bar{\mathbf{W}}_2)$ in its range $\mathcal{R}_\mathcal{M} = \{\boldsymbol{Z} \in \mathbb{R}^{m \times n} \text{ with } \mathrm{rank}(\boldsymbol{Z}) \le \min\{m, k, n\}\}.$

*Proof.* First of all, if $\mathcal{M}(\cdot, \cdot)$ is locally open at $(\bar{\mathbf{W}}_1, \bar{\mathbf{W}}_2)$, according to Proposition 18, the conditions $i)$ and $ii)$ must hold; and hence $\mathrm{rank}(\bar{\mathbf{W}}_1) = \mathrm{rank}(\bar{\mathbf{W}}_2)$ due to Lemma 17. Thus, $\mathcal{M}(\cdot, \cdot)$ cannot be locally open if $\mathrm{rank}(\bar{\mathbf{W}}_1) \ne \mathrm{rank}(\bar{\mathbf{W}}_2)$. On the other hand, when $\mathrm{rank}(\bar{\mathbf{W}}_1) = \mathrm{rank}(\bar{\mathbf{W}}_2)$, the conditions $i)$, $ii)$, $iii)$, and $iv)$ are equivalent due to Lemma 16. Moreover, these conditions imply local openness according to Proposition 18.

$\square$

## C  PROOF OF THEOREM 12

Consider the training problem of a multi-layer deep linear neural network:

$$\underset{\boldsymbol{W}}{\text{minimize}} \quad \frac{1}{2} \| \boldsymbol{W}_h \cdots \boldsymbol{W}_1 \boldsymbol{X} - \boldsymbol{Y} \|^2. \tag{30}$$

Here $\boldsymbol{W} = \left(\boldsymbol{W}_i\right)_{i=1}^h$, $\boldsymbol{W}_i \in \mathbb{R}^{d_i \times d_{i-1}}$ are the weight matrices, $\boldsymbol{X} \in \mathbb{R}^{d_0 \times n}$ is the input training data, and $\boldsymbol{Y} \in \mathbb{R}^{d_h \times n}$ is the target training data. Based on our general framework, the corresponding auxiliary optimization problem is given by

$$\begin{array}{ll} \underset{\boldsymbol{Z} \in \mathbb{R}^{d_h \times d_0}}{\text{minimum}} & \frac{1}{2} \| \boldsymbol{Z}\boldsymbol{X} - \boldsymbol{Y} \|^2 \\ \text{subject to} & \text{rank}(\boldsymbol{Z}) \le d_p \triangleq \min_{0 \le i \le h} d_i \end{array}. \tag{31}$$

Let $p_1^* \triangleq \underset{0 \le i \le h}{\text{argmin}} \, d_i$ and $p_2^* \triangleq \underset{j \ne p_1^*}{\text{argmin}} \, d_j$. In this section we show that if $d_{p_{2*}} < min\{d_h, d_0\}$, we can find a rank deficient $\boldsymbol{Y}$ such that problem (30) has a local minimum that is not global. Otherwise, given any $\boldsymbol{X}$ and $\boldsymbol{Y}$, every local minimum of problem (30) is a global minimum. We start with a Lemma that will be essential in our main proof.

**Lemma 19.** *Consider a degenerate point* $\bar{\mathbf{W}} = (\bar{\mathbf{W}}_h, \ldots, \bar{\mathbf{W}}_1)$ *with* $\mathcal{N}\left(\bar{\mathbf{W}}_i\right)$ *and* $\mathcal{N}\left(\bar{\mathbf{W}}_i^T\right)$ *for* $h - 1 \le i \le 2$ *all non-empty. If*

$$\mathcal{N}\left(\bar{\mathbf{W}}_h\right) \text{ is non-empty} \quad \text{or} \quad \mathcal{N}\left(\bar{\mathbf{W}}_1^T\right) \text{ is non-empty},$$

*then* $\bar{\mathbf{W}}$ *is either a global minimum or a saddle point of problem (30).*

*Proof.* Suppose that $\mathcal{N}\left(\bar{\mathbf{W}}_h\right)$ is non-empty. Let $\boldsymbol{\Delta} = \bar{\mathbf{W}}_h \cdots \bar{\mathbf{W}}_1 \boldsymbol{X} - \boldsymbol{Y}$. If $\boldsymbol{\Delta}\boldsymbol{X}^T = \boldsymbol{0}$, then by convexity of the square loss error function, the point $\bar{\mathbf{W}} = (\bar{\mathbf{W}}_h, \ldots, \bar{\mathbf{W}}_1)$ is a global minimum of (30). Else, there exist $(i, j)$ such that $\left\langle \boldsymbol{X}_{i,:}, \boldsymbol{\Delta}_{j,:} \right\rangle \ne 0$. We define the set $\mathcal{K} \triangleq \{k \,|\, 3 \le k \le h, \quad \mathcal{N}(\bar{\mathbf{W}}_k) \perp \mathcal{N}((\bar{\mathbf{W}}_{k-1}\bar{\mathbf{W}}_{k-2} \cdots \bar{\mathbf{W}}_2)^T)\}$. We split the rest of the proof into two cases that correspond to $\mathcal{K}$ being empty and non-empty.

**Case a:** Assume $\mathcal{K}$ is non-empty. We define $k^* \triangleq \underset{k \in \mathcal{K}}{\text{maximum}} \, k$.

By definition of the set $\mathcal{K}$ and choice of $k^*$, the null space $\mathcal{N}\left(\bar{\mathbf{W}}_{k^*}\right)$ is orthogonal to the null-space $\mathcal{N}\left((\bar{\mathbf{W}}_{k^*-1} \cdots \bar{\mathbf{W}}_2)^T\right)$. This implies there exists a non-zero $\boldsymbol{b} \in \mathbb{R}^{d_{k^*-1}}$ such that $\boldsymbol{b} \in \mathcal{N}\left(\bar{\mathbf{W}}_{k^*}\right) \cap \mathcal{C}\left(\bar{\mathbf{W}}_{k^*-1} \cdots \bar{\mathbf{W}}_2\right)$. By considering perturbation in directions $\boldsymbol{A} = (\boldsymbol{A}_h, \ldots, \boldsymbol{A}_1)$, $\boldsymbol{A}_i \in \mathbb{R}^{d_i \times d_{i-1}}$ for the optimization problem

$$\underset{t}{\text{minimize}} \, g(t) \triangleq \frac{1}{2} \| (\bar{\mathbf{W}}_h + t\boldsymbol{A}_h) \cdots (\bar{\mathbf{W}}_1 + t\boldsymbol{A}_1)\boldsymbol{X} - \boldsymbol{Y} \|^2, \tag{32}$$

we examine the optimality conditions for a specific direction $\bar{\boldsymbol{A}}$.

Let

$$(\bar{\boldsymbol{A}}_h)_{l,:} \triangleq \begin{cases} \alpha_h \boldsymbol{p}_h^T & \text{if } l = j, \\ \boldsymbol{0} & \text{otherwise} \end{cases} \qquad (\bar{\boldsymbol{A}}_1)_{:,l} \triangleq \begin{cases} \alpha_1 \boldsymbol{b}_1 & \text{if } l = i, \\ \boldsymbol{0} & \text{otherwise} \end{cases} \qquad \bar{\boldsymbol{A}}_k \triangleq \begin{cases} \boldsymbol{b}_k \boldsymbol{p}_k^T & \text{if } k^* + 1 \le k \le h - 1 \\ \boldsymbol{b}_k \boldsymbol{b}^T & \text{if } k = k^* \\ \boldsymbol{0} & \text{if } 2 \le k \le k^* - 1, \end{cases}$$

where $\alpha_h$ and $\alpha_1$ are scalar constants, $\boldsymbol{b}_1 \in \mathbb{R}^{d_1}$ such that $\bar{\mathbf{W}}_{k^*-1} \cdots \bar{\mathbf{W}}_2 \boldsymbol{b}_1 = \boldsymbol{b}$, and

$$\boldsymbol{p}_k \in \mathcal{N}\left((\bar{\mathbf{W}}_{k-1} \cdots \bar{\mathbf{W}}_2)^T\right), \quad \boldsymbol{b}_{k-1} \in \mathcal{N}\left(\bar{\mathbf{W}}_k\right), \quad \text{and} \, \langle \boldsymbol{p}_k, \boldsymbol{b}_{k-1} \rangle \ne 0 \quad \forall \, k^* + 1 \le k \le h. \tag{33}$$

Notice that such $\boldsymbol{p}_k$ and $\boldsymbol{b}_{k-1}$ exist from the definition of $\mathcal{K}$ and choice of $k^*$. For this particular choice of $\bar{\boldsymbol{A}} = (\bar{\boldsymbol{A}}_h, \ldots, \bar{\boldsymbol{A}}_1)$, we obtain

$$\bar{\mathbf{W}}_{k+1}\bar{\boldsymbol{A}}_k = \boldsymbol{0} \quad \text{for } k^* \le k \le h - 1 \quad \text{and} \quad \bar{\boldsymbol{A}}_k \bar{\mathbf{W}}_{k-1} \cdots \bar{\mathbf{W}}_2 = \boldsymbol{0} \quad \text{for } k^* + 1 \le k \le h. \tag{34}$$

We now show that $(\bar{\boldsymbol{A}}_h, \ldots, \bar{\boldsymbol{A}}_1)$ is in fact a descent direction. Before proceeding we define some notation that will ease the expressions of the optimality conditions. Let $\mathcal{V}$ be an index set that is a subset of $\{1, \ldots, h\}$. We define the function $f(\bar{\boldsymbol{A}}^{\mathcal{V}}, \bar{\mathbf{W}}^{-\mathcal{V}})$ which is the matrix product attained from $\bar{\mathbf{W}}_h \cdots \bar{\mathbf{W}}_1 \boldsymbol{X}$ by replacing matrices $\bar{\mathbf{W}}_v$ by

matrices $\bar{A}_v$ for every $v \in \mathcal{V}$. For instance, if $h = 5$ and $\mathcal{V} = \{2, 3, 5\}$, then $f(\bar{A}^{\mathcal{V}}, \bar{W}^{-\mathcal{V}}) = \bar{A}_5 \bar{W}_4 \bar{A}_3 \bar{A}_2 \bar{W}_1 X$. We now determine index sets $\mathcal{V}$, with $|\mathcal{V}| \geq 1$, that correspond to non-zero $f(\bar{A}^{\mathcal{V}}, \bar{W}^{-\mathcal{V}})$. First note by definition of $\bar{A}$, if $\mathcal{V} \cap \{k^* - 1, \ldots, 2\} \neq \emptyset$, then $f(\bar{A}^{\mathcal{V}}, \bar{W}^{-\mathcal{V}}) = 0$. Also by (34), for any $k^* \leq v \leq h - 1$, if $v \in \mathcal{V}$ then either $\{k^*, \ldots, h\} \in \mathcal{V}$ or $f(\bar{A}^{\mathcal{V}}, \bar{W}^{-\mathcal{V}}) = 0$. This directly imply that $\bar{A}_h \cdots \bar{A}_{k^*} \bar{W}_{k^*-1} \cdots \bar{W}_1 X$ and $\bar{A}_h \cdots \bar{A}_{k^*} \bar{W}_{k^*-1} \cdots \bar{W}_2 \bar{A}_1 X$ are the only terms that can take non-zero values. Using the definition equation (32) we obtain

$$g(t) = \frac{1}{2} \| t^{h-k^*+1} \bar{A}_h \cdots \bar{A}_{k^*} \bar{W}_{k^*-1} \cdots \bar{W}_1 X + t^{h-k^*+2} \bar{A}_h \cdots \bar{A}_{k^*} \bar{W}_{k^*-1} \cdots \bar{W}_2 \bar{A}_1 X + \Delta \|^2.$$

It follows that

$$\left. \frac{\partial^r g(t)}{\partial t^r} \right|_{t=0} = 0 \quad \text{for all } r \leq h - k^*.$$

and

$$\left. \frac{\partial^{h-k^*+1} g(t)}{\partial t^{h-k^*+1}} \right|_{t=0} = (h - k^* + 1)! \langle \bar{A}_h \cdots \bar{A}_{k^*} \bar{W}_{k^*-1} \cdots \bar{W}_1 X, \Delta \rangle.$$

If $\langle \bar{A}_h \cdots \bar{A}_{k^*} \bar{W}_{k^*-1} \cdots \bar{W}_1 X, \Delta \rangle \neq 0$, then by properly choosing the sign of $\alpha_h$ such that $\langle \bar{A}_h \cdots \bar{A}_{k^*} \bar{W}_{k^*-1} \cdots \bar{W}_1 X, \Delta \rangle < 0$, we get a descent direction. Otherwise, we examine

$$\left. \frac{\partial^{h-k^*+2} g(t)}{\partial t^{h-k^*+2}} \right|_{t=0} = (h - k^* + 2)! \langle \bar{A}_h \cdots \bar{A}_{k^*} \bar{W}_{k^*-1} \cdots \bar{W}_2 \bar{A}_1 X, \Delta \rangle + h(\bar{A}_h \cdots \bar{A}_{k^*} \bar{W}_{k^*-1} \cdots \bar{W}_1 X).$$

where $h(\cdot)$ is a function of $\bar{A}_h \cdots \bar{A}_{k^*} \bar{W}_{k^*-1} \cdots \bar{W}_1 X$.

We now evaluate the term $\langle \bar{A}_h \cdots \bar{A}_{k^*} \bar{W}_{k^*-1} \cdots \bar{W}_2 \bar{A}_1 X, \Delta \rangle$. Since $(\bar{A}_h)_{l,:} = \mathbf{0}$ for all $l \neq j$ and $(\bar{A}_1)_{:,l} = \mathbf{0}$ for all $l \neq i$, we only need to compute the $(j, i)$ index $(\bar{A}_h \cdots \bar{A}_{k^*} \bar{W}_{k^*-1} \cdots \bar{W}_2 \bar{A}_1)_{(j,i)}$ as all other indices are zero. For some constant $c = \boldsymbol{p}_h^T \boldsymbol{b}_{h-1} \boldsymbol{p}_{h-1}^T \boldsymbol{b}_{h-2} \cdots \boldsymbol{p}_{k^*+1}^T \boldsymbol{b}_{k^*} \boldsymbol{b}^T \boldsymbol{b}$, we obtain

$$\left( \bar{A}_h \cdots \bar{A}_{k^*} \bar{W}_{k^*-1} \cdots \bar{W}_2 \bar{A}_1 \right)_{(j,i)} = \alpha_h \alpha_1 \boldsymbol{p}_h^T \boldsymbol{b}_{h-1} \boldsymbol{p}_{h-1}^T \boldsymbol{b}_{h-2} \cdots \boldsymbol{p}_{k^*+1}^T \boldsymbol{b}_{k^*} \boldsymbol{b}^T \bar{W}_{k^*-1} \cdots \bar{W}_2 \boldsymbol{b}_1$$
$$= \alpha_h \alpha_1 \boldsymbol{p}_h^T \boldsymbol{b}_{h-1} \boldsymbol{p}_{h-1}^T \boldsymbol{b}_{h-2} \cdots \boldsymbol{p}_{k^*+1}^T \boldsymbol{b}_{k^*} \boldsymbol{b}^T \boldsymbol{b}$$
$$= \alpha_h \alpha_1 c,$$

where $c$ is non-zero by our choice of $\boldsymbol{b}$, $\boldsymbol{p}_k$ and $\boldsymbol{b}_{k-1}$ for $h \leq k \leq k^* - 1$ as defined in (33). For a fixed $\alpha_h \neq 0$, $h(\bar{A}_h \cdots \bar{A}_{k^*} \bar{W}_{k^*-1} \cdots \bar{W}_1 X)$ is a constant scalar we denote by $c_\alpha$. Then by properly choosing $\alpha_1$ such that

$$\underbrace{\alpha_h}_{\neq 0} \alpha_1 \underbrace{c}_{\neq 0} \underbrace{\langle X_{i,:}, \Delta_{j,:} \rangle}_{\neq 0} + c_\alpha < 0,$$

we get a descent direction. This completes the first case.

**Case b:** Assume $\mathcal{K}$ is empty. We consider

$$(\bar{A}_h)_{l,:} \triangleq \begin{cases} \alpha_h \boldsymbol{p}_h^T & \text{if } l = j, \\ \mathbf{0} & \text{otherwise} \end{cases} \qquad (\bar{A}_1)_{:,l} \triangleq \begin{cases} \alpha_1 \boldsymbol{b}_1 & \text{if } l = i, \\ \mathbf{0} & \text{otherwise} \end{cases} \qquad \bar{A}_k \triangleq \begin{cases} \boldsymbol{b}_k \boldsymbol{p}_k^T & \text{if } 3 \leq k \leq h - 1 \\ \boldsymbol{b}_k \boldsymbol{b}_1^T & \text{if } k = 2, \end{cases}$$

where $\alpha_h$ and $\alpha_1$ are scalar constants, $\boldsymbol{b}_1 \in \mathcal{N}(\bar{W}_2)$, and

$$\boldsymbol{p}_k \in \mathcal{N}\left( (\bar{W}_{k-1} \cdots \bar{W}_2)^T \right), \quad \boldsymbol{b}_{k-1} \in \mathcal{N}(\bar{W}_k), \quad \text{and } \langle \boldsymbol{p}_k, \boldsymbol{b}_{k-1} \rangle \neq 0 \quad \forall \, 3 \leq k \leq h. \tag{35}$$

For this particular choice of $\bar{A} = (\bar{A}_h, \ldots, \bar{A}_1)$, we obtain

$$\bar{W}_{k+1} \bar{A}_k = 0 \quad \text{for } 2 \leq k \leq h - 1 \quad \text{and} \quad \bar{A}_k \bar{W}_{k-1} \cdots \bar{W}_2 = 0 \quad \text{for } 3 \leq k \leq h.$$

We now determine index sets $\mathcal{V}$, with $|\mathcal{V}| \geq 1$, that correspond to non-zero $f(\bar{\boldsymbol{A}}^{\mathcal{V}}, \bar{\mathbf{W}}^{-\mathcal{V}})$. By (35), for any $2 \leq v \leq h - 1$, if $v \in \mathcal{V}$ then either $\{2, \ldots, h\} \in \mathcal{V}$ or $f(\bar{\boldsymbol{A}}^{\mathcal{V}}, \bar{\mathbf{W}}^{-\mathcal{V}}) = 0$. This directly imply that $\bar{\boldsymbol{A}}_h \cdots \bar{\boldsymbol{A}}_2 \bar{\mathbf{W}}_1 \boldsymbol{X}$ and $\bar{\boldsymbol{A}}_h \cdots \bar{\boldsymbol{A}}_1 \boldsymbol{X}$ are the only terms that can take non-zero values. Using the definition of equation (32) we obtain

$$g(t) = \frac{1}{2}\|t^{h-1}\bar{\boldsymbol{A}}_h \cdots \bar{\boldsymbol{A}}_2 \bar{\mathbf{W}}_1 \boldsymbol{X} + t^h \bar{\boldsymbol{A}}_h \cdots \bar{\boldsymbol{A}}_1 \boldsymbol{X} + \boldsymbol{\Delta}\|^2.$$

It follows that

$$\left.\frac{\partial^r g(t)}{\partial t^r}\right|_{t=0} = 0 \quad \text{for all } r \leq h - 2.$$

and

$$\left.\frac{\partial^{h-1} g(t)}{\partial t^{h-1}}\right|_{t=0} = (h-1)!\big\langle \bar{\boldsymbol{A}}_h \cdots \bar{\boldsymbol{A}}_2 \bar{\mathbf{W}}_1 \boldsymbol{X}, \boldsymbol{\Delta} \big\rangle.$$

If $\big\langle \bar{\boldsymbol{A}}_h \cdots \bar{\boldsymbol{A}}_2 \bar{\mathbf{W}}_1 \boldsymbol{X}, \boldsymbol{\Delta} \big\rangle \neq 0$, then by properly choosing the sign of $\alpha_h$ such that $\big\langle \bar{\boldsymbol{A}}_h \cdots \bar{\boldsymbol{A}}_2 \bar{\mathbf{W}}_1 \boldsymbol{X}, \boldsymbol{\Delta} \big\rangle < 0$, we get a descent direction. Otherwise, we examine

$$\left.\frac{\partial^h g(t)}{\partial t^h}\right|_{t=0} = \big\langle \bar{\boldsymbol{A}}_h \cdots \bar{\boldsymbol{A}}_1 \boldsymbol{X}, \boldsymbol{\Delta} \big\rangle + h(\bar{\boldsymbol{A}}_h \cdots \bar{\boldsymbol{A}}_2 \bar{\mathbf{W}}_1 \boldsymbol{X}),$$

where $h(\cdot)$ is a function of $\bar{\boldsymbol{A}}_h \cdots \bar{\boldsymbol{A}}_2 \bar{\mathbf{W}}_1 \boldsymbol{X}$. We now evaluate the term $\big\langle \bar{\boldsymbol{A}}_h \cdots \bar{\boldsymbol{A}}_1 \boldsymbol{X}, \boldsymbol{\Delta} \big\rangle$. Since $(\bar{\boldsymbol{A}}_h)_{l,:} = \mathbf{0}$ for all $l \neq j$ and $(\bar{\boldsymbol{A}}_1)_{:,l} = \mathbf{0}$ for all $l \neq i$, we only need to compute the $(j, i)$ index $(\bar{\boldsymbol{A}}_h \cdots \bar{\boldsymbol{A}}_1)_{(j,i)}$ as all other indices are zero. For some constant $c = \boldsymbol{p}_h^T \boldsymbol{b}_{h-1} \boldsymbol{p}_{h-1}^T \boldsymbol{b}_{h-2} \cdots \boldsymbol{p}_3^T \boldsymbol{b}_2 \boldsymbol{b}_1^T \boldsymbol{b}_1$, we obtain

$$\begin{aligned}
\big(\bar{\boldsymbol{A}}_h \cdots \bar{\boldsymbol{A}}_1\big)_{(j,i)} &= \alpha_h \alpha_1 \boldsymbol{p}_h^T \boldsymbol{b}_{h-1} \boldsymbol{p}_{h-1}^T \boldsymbol{b}_{h-2} \cdots \boldsymbol{p}_3^T \boldsymbol{b}_2 \boldsymbol{b}_1^T \boldsymbol{b}_1 \\
&= \alpha_h \alpha_1 c,
\end{aligned}$$

where $c$ is non-zero by our choice of $\boldsymbol{b}$, $\boldsymbol{p}_k$ and $\boldsymbol{b}_{k-1}$ for $3 \leq k \leq h$ as defined in (35). For a fixed $\alpha_h \neq 0$, $h(\bar{\boldsymbol{A}}_h \cdots \bar{\boldsymbol{A}}_2 \bar{\mathbf{W}}_1 \boldsymbol{X})$ is a constant scalar we denote by $c_\alpha$. Then by properly choosing $\alpha_1$ such that

$$\underbrace{\alpha_h}_{\neq 0} \alpha_1 \underbrace{c}_{\neq 0} \underbrace{\big\langle \boldsymbol{X}_{i,:}, \boldsymbol{\Delta}_{j,:} \big\rangle}_{\neq 0} + c_\alpha < 0,$$

we get a descent direction. This completes the second case.

Now if $\mathcal{N}\big(\bar{\mathbf{W}}_1^T\big)$ is non-empty, we define the set

$$\mathcal{K} \triangleq \{k \,|\, 1 \leq k \leq h - 2, \ \mathcal{N}(\bar{\mathbf{W}}_{h-1} \cdots \bar{\mathbf{W}}_{k+1}) \perp \mathcal{N}(\bar{\mathbf{W}}_k^T)\},$$

and use a similar proof scheme to show the result. More specifically, we split the proof into two cases that correspond to $\mathcal{K}$ being empty and non-empty.

**Case a:** Assume $\mathcal{K}$ is non-empty. We define $k^* \triangleq \underset{k \in \mathcal{K}}{\text{minimum }} k$.

By definition of the set $\mathcal{K}$ and choice of $k^*$, the null space $\mathcal{N}\big(\bar{\mathbf{W}}_{k^*}^T\big)$ is orthogonal to the null-space $\mathcal{N}\big(\bar{\mathbf{W}}_{h-1} \cdots \bar{\mathbf{W}}_{k^*+1}\big)$. This implies there exists a non-zero $\boldsymbol{p} \in \mathbb{R}^{d_{k^*}}$ such that $\boldsymbol{p} \in \mathcal{N}\big(\bar{\mathbf{W}}_{k^*}^T\big) \cap \mathcal{C}\big((\bar{\mathbf{W}}_{h-1} \cdots \bar{\mathbf{W}}_{k^*+1})^T\big)$. By considering perturbation in directions $\boldsymbol{A} = (\boldsymbol{A}_h, \ldots, \boldsymbol{A}_1)$, $\boldsymbol{A}_i \in \mathbb{R}^{d_i \times d_{i-1}}$ for the optimization problem

$$\underset{t}{\text{minimize }} g(t) \triangleq \frac{1}{2}\|(\bar{\mathbf{W}}_h + t\boldsymbol{A}_h) \cdots (\bar{\mathbf{W}}_1 + t\boldsymbol{A}_1)\boldsymbol{X} - \boldsymbol{Y}\|^2, \tag{36}$$

we examine the optimality conditions for a specific direction $\bar{\boldsymbol{A}}$.

Let

$$(\bar{\boldsymbol{A}}_h)_{l,:} \triangleq \begin{cases} \alpha_h \boldsymbol{p}_h^T & \text{if } l = j, \\ \boldsymbol{0} & \text{otherwise} \end{cases} \qquad (\bar{\boldsymbol{A}}_1)_{:,l} \triangleq \begin{cases} \alpha_1 \boldsymbol{b}_1 & \text{if } l = i, \\ \boldsymbol{0} & \text{otherwise} \end{cases} \qquad \bar{\boldsymbol{A}}_k \triangleq \begin{cases} \boldsymbol{b}_k \boldsymbol{p}_k^T & \text{if } 2 \leq k \leq k^* - 1 \\ \boldsymbol{p} \boldsymbol{p}_k^T & \text{if } k = k^* \\ \boldsymbol{0} & \text{if } k^* + 1 \leq k \leq h - 1, \end{cases}$$

where $\alpha_h$ and $\alpha_1$ are scalar constants, $\boldsymbol{p}_h \in \mathbb{R}^{d_h-1}$ such that $\boldsymbol{p}_h^T \bar{\mathbf{W}}_{h-1} \cdots \bar{\mathbf{W}}_{k^*+1} = \boldsymbol{p}^T$, and

$$\boldsymbol{p}_k \in \mathcal{N}(\bar{\mathbf{W}}_{k-1}^T), \quad \boldsymbol{b}_{k-1} \in \mathcal{N}(\bar{\mathbf{W}}_{h-1} \cdots \bar{\mathbf{W}}_k), \quad \text{and } \langle \boldsymbol{p}_k, \boldsymbol{b}_{k-1} \rangle \neq 0 \quad \forall\, 2 \leq k \leq k^*. \tag{37}$$

Notice that such $\boldsymbol{p}_k$ and $\boldsymbol{b}_{k-1}$ exist from the definition of $\mathcal{K}$ and choice of $k^*$. For this particular choice of $\bar{\boldsymbol{A}} = (\bar{\boldsymbol{A}}_h, \ldots, \bar{\boldsymbol{A}}_1)$, we obtain

$$\bar{\boldsymbol{A}}_k \bar{\mathbf{W}}_{k-1} = \boldsymbol{0} \quad \text{for } 2 \leq k \leq k^* \quad \text{and} \quad \bar{\mathbf{W}}_{h-1} \cdots \bar{\mathbf{W}}_{k+1} \bar{\boldsymbol{A}}_k = \boldsymbol{0} \quad \text{for } 1 \leq k \leq k^* - 1. \tag{38}$$

The same argument used above can be used to show that $(\bar{\boldsymbol{A}}_h, \ldots, \bar{\boldsymbol{A}}_1)$ is actually a descent direction. This completes the first case.

**Case b:** Assume $\mathcal{K}$ is empty. We consider

$$(\bar{\boldsymbol{A}}_h)_{l,:} \triangleq \begin{cases} \alpha_h \boldsymbol{p}_h^T & \text{if } l = j, \\ \boldsymbol{0} & \text{otherwise} \end{cases} \qquad (\bar{\boldsymbol{A}}_1)_{:,l} \triangleq \begin{cases} \alpha_1 \boldsymbol{b}_1 & \text{if } l = i, \\ \boldsymbol{0} & \text{otherwise} \end{cases} \qquad \bar{\boldsymbol{A}}_k \triangleq \begin{cases} \boldsymbol{b}_k \boldsymbol{p}_k^T & \text{if } 2 \leq k \leq h - 2 \\ \boldsymbol{p}_h \boldsymbol{p}_k^T & \text{if } k = h - 1, \end{cases}$$

where $\alpha_h$ and $\alpha_1$ are scalar constants, $\boldsymbol{p}_h \in \mathcal{N}(\bar{\mathbf{W}}_{h-1}^T)$, and

$$\boldsymbol{p}_k \in \mathcal{N}(\bar{\mathbf{W}}_{k-1}^T), \quad \boldsymbol{b}_{k-1} \in \mathcal{N}(\bar{\mathbf{W}}_{h-1} \cdots \bar{\mathbf{W}}_k), \quad \text{and } \langle \boldsymbol{p}_k, \boldsymbol{b}_{k-1} \rangle \neq 0 \quad \forall\, 2 \leq k \leq h - 1. \tag{39}$$

For this particular choice of $\bar{\boldsymbol{A}} = (\bar{\boldsymbol{A}}_h, \ldots, \bar{\boldsymbol{A}}_1)$, we obtain

$$\bar{\boldsymbol{A}}_k \bar{\mathbf{W}}_{k-1} = \boldsymbol{0} \quad \text{for } 2 \leq k \leq h - 1 \quad \text{and} \quad \bar{\mathbf{W}}_{h-1} \cdots \bar{\mathbf{W}}_{k+1} \bar{\boldsymbol{A}}_k = \boldsymbol{0} \quad \text{for } 1 \leq k \leq h - 2. \tag{40}$$

The same argument used above can be used to show that $(\bar{\boldsymbol{A}}_h, \ldots, \bar{\boldsymbol{A}}_1)$ is actually a descent direction. This completes the first case and thus completes the proof. $\qquad \square$

Note that following the same steps of the proof in Lemma 19, we get the same result when replacing the square loss error by a general convex and differentiable function $\ell(\cdot)$. We are now ready to prove the main result restated below.

**Theorem** 12 Let $p_1^* \triangleq \underset{0 \leq i \leq h}{\operatorname{argmin}} d_i$ and $p_2^* \triangleq \underset{j \neq p_1^*}{\operatorname{argmin}} d_j$. If $d_{p_{2*}} < min(d_h, d_0)$, we can find a rank deficient $\boldsymbol{Y}$ such that problem (30) has a local minimum that is not global. Otherwise, given any $\boldsymbol{X}$ and $\boldsymbol{Y}$, every local minimum of problem (30) is a global minimum.

*Proof.* Suppose $d_{p_2^*} < min\{d_h, d_0\}$, we define

$$p_2 \triangleq \max(p_1^*, p_2^*) \quad \text{and} \quad p_1 \triangleq \min(p_1^*, p_2^*).$$

Let $\boldsymbol{X} \triangleq \boldsymbol{I}$,

$$(\bar{\boldsymbol{Y}})_{(i,j)} \triangleq \begin{cases} 1 & \text{if } (i, j) = (d_h, d_0) \\ 0 & \text{otherwise} \end{cases}, \qquad \bar{\mathbf{W}}_k \triangleq \begin{cases} \begin{bmatrix} \boldsymbol{I}_{d_k} & \boldsymbol{0} \end{bmatrix} & \text{if } d_k \leq d_{k-1}, \\ \begin{bmatrix} \boldsymbol{I}_{d_{k-1}} \\ \boldsymbol{0} \end{bmatrix} & \text{if } d_k > d_{k-1}, \end{cases}$$

for $k \in \{h, \ldots, p_2 + 1\} \cup \{p_1, \ldots, 1\}$, and $\bar{\mathbf{W}}_k = \boldsymbol{0}$ for $k \in \{p_2, \ldots, p_1 + 1\}$. Since $\bar{\boldsymbol{W}}_h \cdots \bar{\boldsymbol{W}}_{p_2+1}$ and $\bar{\boldsymbol{W}}_{p_1} \cdots \bar{\boldsymbol{W}}_1$ are both full rank, then using Lemma 10, the matrix products $\mathcal{M}_{h,p_2+1}$ and $\mathcal{M}_{p_1,1}$ are locally open at $(\bar{\mathbf{W}}_h, \ldots, \bar{\mathbf{W}}_{p_2+1})$ and $(\bar{\mathbf{W}}_{p_1}, \ldots, \bar{\mathbf{W}}_1)$, respectively. Moreover, using Proposition 4 and the composition property

of open maps, the matrix product mapping $\mathcal{M}_{p_2,p_1+1}$ is locally open at $(\bar{\mathbf{W}}_{p_2}, \ldots, \bar{\mathbf{W}}_{p_1+1})$. It follows by Observation 1 that if $\bar{\mathbf{W}}$ is a local minimum of

$$\underset{\mathbf{W}}{\text{minimize}} \; \frac{1}{2}\|\boldsymbol{W}_h \cdots \boldsymbol{W}_1 - \bar{\boldsymbol{Y}}\|^2. \tag{41}$$

then $(\bar{\boldsymbol{Z}}_3, \bar{\boldsymbol{Z}}_2, \bar{\boldsymbol{Z}}_1)$ is a local minimum of

$$\underset{\boldsymbol{Z}_3 \in \mathbb{R}^{d_h \times d_{p_2}}, \, \boldsymbol{Z}_2 \in \mathbb{R}^{d_{p_2} \times d_{p_1}}, \, \boldsymbol{Z}_1 \in \mathbb{R}^{d_{p_1} \times d_0}}{\text{minimize}} \; \frac{1}{2}\|\boldsymbol{Z}_3 \boldsymbol{Z}_2 \boldsymbol{Z}_1 - \bar{\boldsymbol{Y}}\|^2. \tag{42}$$

where

$$\bar{\boldsymbol{Z}}_3 = \bar{\mathbf{W}}_h \cdots \bar{\mathbf{W}}_{p_2+1} = \begin{bmatrix} \boldsymbol{I}_{d_{p_2}} \\ \boldsymbol{0} \end{bmatrix}, \quad \bar{\boldsymbol{Z}}_2 = \boldsymbol{0}, \quad \text{and} \quad \bar{\boldsymbol{Z}}_1 = \bar{\mathbf{W}}_{p_1} \cdots \bar{\mathbf{W}}_1 = \begin{bmatrix} \boldsymbol{I}_{d_{p_1}} & \boldsymbol{0} \end{bmatrix}.$$

The point $(\bar{\boldsymbol{Z}}_3, \bar{\boldsymbol{Z}}_2, \bar{\boldsymbol{Z}}_1)$ is obviously not global, we show using optimality conditions that the point is a local minimum. By considering perturbations in the directions $\bar{\boldsymbol{A}} = (\bar{\boldsymbol{A}}_3, \bar{\boldsymbol{A}}_2, \bar{\boldsymbol{A}}_1)$ for the optimization problem

$$\underset{t}{\text{minimize}} \; g(t) \; \triangleq \frac{1}{2}\|(\bar{\boldsymbol{Z}}_3 + t\bar{\boldsymbol{A}}_3)(\bar{\boldsymbol{Z}}_2 + t\bar{\boldsymbol{A}}_2)(\bar{\boldsymbol{Z}}_1 + t\bar{\boldsymbol{A}}_1) - \bar{\boldsymbol{Y}}\|^2$$

$$\tag{43}$$

$$= \frac{1}{2}\|t(\bar{\boldsymbol{Z}}_3 + t\bar{\boldsymbol{A}}_3)\bar{\boldsymbol{A}}_2(\bar{\boldsymbol{Z}}_1 + t\bar{\boldsymbol{A}}_1) - \bar{\boldsymbol{Y}}\|^2.$$

It follows that

$$\begin{aligned} \left.\frac{\partial g(t)}{\partial t}\right|_{t=0} &= \big\langle \bar{\boldsymbol{Z}}_3 \bar{\boldsymbol{A}}_2 \bar{\boldsymbol{Z}}_1, -\bar{\boldsymbol{Y}} \big\rangle \\ &= -\big(\bar{\boldsymbol{Z}}_3 \bar{\boldsymbol{A}}_2 \bar{\boldsymbol{Z}}_1\big)_{d_h,d_0} \bar{\boldsymbol{Y}}_{d_h,d_0} \\ &= 0, \end{aligned} \tag{44}$$

where the last equality holds since the last row ($d_h^{th}$ row) of $\bar{\boldsymbol{Z}}_3$ is zero. Also,

$$\begin{aligned} \left.\frac{\partial^2 g(t)}{\partial t^2}\right|_{t=0} &= 2\big\langle \bar{\boldsymbol{Z}}_3 \bar{\boldsymbol{A}}_2 \bar{\boldsymbol{A}}_1 + \bar{\boldsymbol{A}}_3 \bar{\boldsymbol{A}}_2 \bar{\boldsymbol{Z}}_1, -\bar{\boldsymbol{Y}} \big\rangle + \|\bar{\boldsymbol{Z}}_3 \bar{\boldsymbol{A}}_2 \bar{\boldsymbol{Z}}_1\|^2 \\ &= -2\big(\bar{\boldsymbol{Z}}_3 \bar{\boldsymbol{A}}_2 \bar{\boldsymbol{A}}_1\big)_{d_h,d_0} \bar{\boldsymbol{Y}}_{d_h,d_0} - 2\big(\bar{\boldsymbol{A}}_3 \bar{\boldsymbol{A}}_2 \bar{\boldsymbol{Z}}_1\big)_{d_h,d_0} \bar{\boldsymbol{Y}}_{d_h,d_0} + \|\bar{\boldsymbol{Z}}_3 \bar{\boldsymbol{A}}_2 \bar{\boldsymbol{Z}}_1\|^2 \\ &= \|\bar{\boldsymbol{A}}_2\|^2, \end{aligned} \tag{45}$$

where the last equality holds since the last row ($d_h^{th}$ row) of $\bar{\boldsymbol{Z}}_3$ and the last column ($d_0^{th}$ column) of $\bar{\boldsymbol{Z}}_1$ are both zeros. Then for $\|\bar{\boldsymbol{A}}_2\| \neq 0$, it follows from the second-order optimalty condition that the point is a local minimum, and if $\|\bar{\boldsymbol{A}}_2\| = 0$ we get

$$g(t) = \frac{1}{2}\|\bar{\boldsymbol{Y}}\| = \frac{1}{2}$$

which implies $(\bar{\boldsymbol{Z}}_3, \bar{\boldsymbol{Z}}_2, \bar{\boldsymbol{Z}}_1)$ is a local optimum that is not global.

Note that the same method used to construct the example above can be used to find a local minimum that is not global whenever the $\text{rank}(\boldsymbol{Y}) \leq \min\{d_h - d_{p_2}, d_0 - d_{p_1}\}$. When $\boldsymbol{Y}$ is full rank, we know from the results of Lu & Kawaguchi (2017); Yun et al. (2017) that every local minimum is global. To have a complete characterization of problems for which every local minimum is global, it remains to either prove or disprove the statement when $\boldsymbol{Y}$ is a rank deficient matrix with $\text{rank}(\boldsymbol{Y}) > \min\{d_h - d_{p_2}, d_0 - d_{p_1}\}$. We now provide a counterexample that disproves the statement. In particular, we construct a three layer network with input $\boldsymbol{X}$ and output $\boldsymbol{Y}$ with $\text{rank}(\boldsymbol{Y}) > \min\{d_h - d_{p_2}, d_0 - d_{p_1}\}$, and then find a local minimum $(\bar{\mathbf{W}}_3, \bar{\mathbf{W}}_2, \bar{\mathbf{W}}_1)$ that is not global. Let $\boldsymbol{X} = \boldsymbol{I}$,

$$\boldsymbol{Y} \triangleq \begin{bmatrix} 1 & 0 & -1 \\ 0 & 4 & 0 \\ -1 & 0 & 1 \end{bmatrix}, \quad \bar{\mathbf{W}}_3 \triangleq \begin{bmatrix} 1 & -1 \\ -1 & -1 \\ 1 & -1 \end{bmatrix}, \quad \bar{\mathbf{W}}_2 \triangleq \begin{bmatrix} 1 & 1 \\ 1 & 1 \end{bmatrix}, \quad \text{and} \quad \bar{\mathbf{W}}_1 \triangleq \bar{\mathbf{W}}_3^T.$$

Obviously $(\bar{\mathbf{W}}_3, \bar{\mathbf{W}}_2, \bar{\mathbf{W}}_1)$ is not a global minimum. We define $\boldsymbol{\Delta} \triangleq \bar{\mathbf{W}}_3 \bar{\mathbf{W}}_2 \bar{\mathbf{W}}_1 - \boldsymbol{Y}$. Then we get

$$\bar{\mathbf{W}}_3^T \boldsymbol{\Delta} = \boldsymbol{\Delta} \boldsymbol{W}_1^T = \boldsymbol{0}. \tag{46}$$

By considering perturbations in the directions $\boldsymbol{A} = (\boldsymbol{A}_3, \boldsymbol{A}_2, \boldsymbol{A}_1)$ for the optimization problem

$$\underset{t}{\text{minimize}} \; g(t, \boldsymbol{A}) \triangleq \frac{1}{2} \|(\bar{\mathbf{W}}_3 + t\boldsymbol{A}_3)(\bar{\mathbf{W}}_3 + t\boldsymbol{A}_2)(\bar{\mathbf{W}}_1 + t\boldsymbol{A}_1) - \boldsymbol{Y}\|^2,$$

it follows that

$$\left. \frac{\partial g(t, \boldsymbol{A})}{\partial t} \right|_{t=0} = \left\langle \bar{\mathbf{W}}_3 \bar{\mathbf{W}}_2 \boldsymbol{A}_1 + \bar{\mathbf{W}}_3 \boldsymbol{A}_2 \bar{\mathbf{W}}_1 + \boldsymbol{A}_3 \bar{\mathbf{W}}_2 \bar{\mathbf{W}}_1, \boldsymbol{\Delta} \right\rangle = 0,$$

where the last equality is directly implied from (46). Also

$$g^{(2)}(0, \boldsymbol{A}) \triangleq \left. \frac{\partial^2 g(t, \boldsymbol{A})}{\partial t^2} \right|_{t=0} = 2\left\langle \boldsymbol{A}_3 \boldsymbol{A}_2 \bar{\mathbf{W}}_1 + \boldsymbol{A}_3 \bar{\mathbf{W}}_2 \boldsymbol{A}_1 + \bar{\mathbf{W}}_3 \boldsymbol{A}_2 \boldsymbol{A}_1, \boldsymbol{\Delta} \right\rangle + \|\bar{\mathbf{W}}_3 \bar{\mathbf{W}}_2 \boldsymbol{A}_1 + \bar{\mathbf{W}}_3 \boldsymbol{A}_2 \bar{\mathbf{W}}_1 + \boldsymbol{A}_3 \bar{\mathbf{W}}_2 \bar{\mathbf{W}}_1 \|^2$$

$$= 2\left\langle \boldsymbol{A}_3 \bar{\mathbf{W}}_2 \boldsymbol{A}_1, \boldsymbol{\Delta} \right\rangle + \|\bar{\mathbf{W}}_3 \bar{\mathbf{W}}_2 \boldsymbol{A}_1 + \bar{\mathbf{W}}_3 \boldsymbol{A}_2 \bar{\mathbf{W}}_1 + \boldsymbol{A}_3 \bar{\mathbf{W}}_2 \bar{\mathbf{W}}_1 \|^2.$$

which is a quadratic function of $\boldsymbol{A}$ we denote

$$f_{\boldsymbol{A}} \triangleq \frac{1}{2} \boldsymbol{a}^T \boldsymbol{H}_{\boldsymbol{A}} \boldsymbol{a}.$$

Here $\boldsymbol{a} \in \mathbb{R}^{16 \times 1}$ is a vectorization of matrices $\boldsymbol{A}_3$, $\boldsymbol{A}_2$, and $\boldsymbol{A}_1$, and $\boldsymbol{H}_{\boldsymbol{A}}$ is the hessian of $f_{\boldsymbol{A}}$. By computing the eigenvalues of $\boldsymbol{H}_{\boldsymbol{A}}$ we get that $\boldsymbol{H}_{\boldsymbol{A}} \succeq 0$ which directly implies

$$g^{(2)}(0, \boldsymbol{A}) \geq 0 \quad \forall \boldsymbol{A}.$$

Moreover, let $\boldsymbol{a}_{\text{opt}}$ be the optimal solution set of the problem

$$\underset{\boldsymbol{a}}{\text{minimize}} \; f_{\boldsymbol{A}}.$$

Then $\boldsymbol{a}_{\text{opt}} = \{\boldsymbol{a} \,|\, \boldsymbol{a} \in \mathcal{N}(\boldsymbol{H}_{\boldsymbol{a}}\}$. We notice that for any $\bar{\boldsymbol{a}} \in \boldsymbol{a}_{\text{opt}}$, the corresponding direction $\bar{\boldsymbol{A}}$ has

$$\bar{\mathbf{W}}_3 \bar{\mathbf{W}}_2 \bar{\boldsymbol{A}}_1 + \bar{\mathbf{W}}_3 \bar{\boldsymbol{A}}_2 \bar{\mathbf{W}}_1 + \bar{\boldsymbol{A}}_3 \bar{\mathbf{W}}_2 \bar{\mathbf{W}}_1 = \boldsymbol{0} \quad \text{and} \quad \left\langle \bar{\boldsymbol{A}}_3 \bar{\boldsymbol{A}}_2 \bar{\boldsymbol{A}}_1, \boldsymbol{\Delta} \right\rangle = 0.$$

Then, it follows that

$$g^{(3)}(0, \bar{\boldsymbol{A}}) \triangleq \left. \frac{\partial^3 g(t, \bar{\boldsymbol{A}})}{\partial t^3} \right|_{t=0} = 6\left\langle \bar{\boldsymbol{A}}_3 \bar{\boldsymbol{A}}_2 \bar{\boldsymbol{A}}_1, \boldsymbol{\Delta} \right\rangle = 0,$$

and

$$g^{(4)}(0, \bar{\boldsymbol{A}}) \triangleq \left. \frac{\partial^4 g(t, \bar{\boldsymbol{A}})}{\partial t^4} \right|_{t=0} = 12\|\bar{\boldsymbol{A}}_3 \bar{\boldsymbol{A}}_2 \bar{\mathbf{W}}_1 + \bar{\boldsymbol{A}}_3 \bar{\mathbf{W}}_2 \bar{\boldsymbol{A}}_1 + \bar{\mathbf{W}}_3 \bar{\boldsymbol{A}}_2 \bar{\boldsymbol{A}}_1 \|^2 \geq 0.$$

If $\bar{\boldsymbol{A}}_3 \bar{\boldsymbol{A}}_2 \bar{\mathbf{W}}_1 + \bar{\boldsymbol{A}}_3 \bar{\mathbf{W}}_2 \bar{\boldsymbol{A}}_1 + \bar{\mathbf{W}}_3 \bar{\boldsymbol{A}}_2 \bar{\boldsymbol{A}}_1 \neq \boldsymbol{0}$, then using the fourth order optimality conditions $(\bar{\mathbf{W}}_3, \bar{\mathbf{W}}_2, \bar{\mathbf{W}}_1)$ is a local minimum. Otherwise, we get

$$g^{(5)}(0, \bar{\boldsymbol{A}}) \triangleq \left. \frac{\partial^5 g(t, \bar{\boldsymbol{A}})}{\partial t^5} \right|_{t=0} = 0,$$

and

$$g^{(6)}(0, \bar{\boldsymbol{A}}) \triangleq \left. \frac{\partial^5 g(t, \bar{\boldsymbol{A}})}{\partial t^5} \right|_{t=0} = \|\bar{\boldsymbol{A}}_3 \bar{\boldsymbol{A}}_2 \bar{\boldsymbol{A}}_1\|^2 \geq 0,$$

which also implies that $(\bar{\mathbf{W}}_3, \bar{\mathbf{W}}_2, \bar{\mathbf{W}}_1)$ is a local minimum.

We now show that if $d_{p_2^*} \geq min\{d_h, d_0\}$, every local minimum of (30) is global. In particular, we show that for any $\boldsymbol{X}$ and $\boldsymbol{Y}$, if $\bar{\mathbf{W}}$ is not a global minimum, we can construct a descent direction.

First notice that if for some $1 \leq i \leq h-1$, $\bar{\mathbf{W}}_i$ is full column rank, then using Proposition 4, $\mathcal{M}_{i+1,i}(\cdot)$ is locally open at $(\bar{\mathbf{W}}_{i+1}, \bar{\mathbf{W}}_i)$ and $\bar{\mathbf{W}}_{i+1}\bar{\mathbf{W}}_i \in \mathbb{R}^{d_{i+1} \times d_{i-1}}$. Using Observation 1, we conclude that any local minimum of problem (30) is a local minimum of the problem obtained by replacing $\bar{\mathbf{W}}_{i+1}\bar{\mathbf{W}}_i$ by $\bar{\boldsymbol{Z}}_{i+1,i} \in \mathbb{R}^{d_{i+1} \times d_{i-1}}$. By a similar

argument, we conclude that if $\bar{\mathbf{W}}_i$ is a full row rank for some $2 \leq i \leq h$, any local minimum of problem (30) is a local minimum of the problem obtained by replacing $\bar{\mathbf{W}}_i \bar{\mathbf{W}}_{i-1}$ by $\bar{\mathbf{Z}}_{i,i-1} \in \mathbb{R}^{d_i \times d_{i-2}}$. Thus, if $\bar{\mathbf{W}} = (\bar{\mathbf{W}}_h, \ldots, \bar{\mathbf{W}}_1)$ is a local minimum of problem (30), the new point $\bar{\mathbf{Z}} = (\bar{\mathbf{Z}}'_{h'}, \ldots, \bar{\mathbf{Z}}'_1)$, where $\bar{\mathbf{Z}}_i \in \mathbb{R}^{d'_i \times d'_{i-1}}$ and $h' \leq h$, is a local minimum of the problem attained by applying the replacements discussed above. If $h' = 1$, we get the desired result from Lemma 7. Else, if $h' = 2$, the auxiliary problem becomes a two layer linear network for which Theorem 8 provides the desired result. When $h' > 2$, examine $d'_{h'}, d'_{h'-1}, d'_1$ and $d'_0$. If $d'_{h'} > d'_{h'-1}$ and $d'_0 > d'_1$, then $d_{p_2^*} < min\{d_h, d_0\}$ which contradicts our assumption. It follows by construction of $\bar{\mathbf{Z}}_i$, that either $d'_{h'} \leq d'_{h'-1}$ and $\bar{\mathbf{Z}}'_{h'}$ is not full row rank or $d'_0 \leq d'_1$ and $\bar{\mathbf{Z}}'_1$ is not full column rank; thus at least one of the null spaces $\mathcal{N}\big((\bar{\mathbf{Z}}'_{h'})^T\big)$, $\mathcal{N}\big(\bar{\mathbf{Z}}'_1\big)$ is non empty. Moreover, $\bar{\mathbf{Z}}_i$ has non-empty right and left null spaces for $2 \leq i \leq h - 1$. The result follows using Lemma 19.

$\square$

