# OpenReview forum: "Learning Deep Models: Critical Points and Local Openness"
_ICLR.cc/2018/Conference — Invite to Workshop Track_

### Official Review · AnonReviewer1 · 2017-11-18
**Nice presentation, useful results, interesting future directions. Weak accept mostly for the reason that the paper mostly reproduces similar results in the literature with a different methodology. I'm not strongly opinioned until I see also the other reviews.**

**Rating:** 6
**Confidence:** 4

**Review:**

Summary: The paper focuses on the characterization of the landscape of deep neural networks; i.e., when and why local minima are global, what are the conditions for saddle critical points, etc. The paper covers a somewhat wide range of deep nets (from shallow with linear activation to deeper with non-linear activation); it focuces only on feed forward neural networks.
As the authors state, this paper provides a unifying perspective to the subject (it justifies the results of others through this unifying theory, but also provides new results; e.g., there are results that do not depend on assumptions on the target data matrix Y).

Originality: The paper provides similar results to previous work, while removing some of the assumptions made in previous work. In that sense, the originality of the results is weak, but definitely there is some novelty in the methodology used to get to these results. Thus, I would say original.

Importance: The paper deals with the important problem of when and why training algorithms might get to global/local/saddle critical points. While there are no direct connections with generalization properties, characterizing the landscape of neural networks is an important topic to make further steps into better understanding of deep learning. It will attract some attention at the conference.

Clarity: The paper is well-written - some parts need improvement, but overall I'm satisfied with the current version.

Comments:
1. If problem (4) is not considered at all in this paper (in its full generality that considers matrix completion and matrix sensing as special cases), then the authors could just start with the model in (5).

2. Remark 1 has a nice example - could this example be shown with Y not being the all-zeros vector?

3. In section 5, the authors make a connection with the work of Ge et al. 2016. They state that the problems in (10)-(11) constitute generalizations of the symmetric matrix completion case, considered in Ge et al. 2016. However, in that work, the main difficulty of proving global optimality comes from the randomness of the sampling mask operator (which introduces the notion of incoherence and requires results in expectation). It is not clear, and maybe it is an overstatement, that the results in section 5 generalize that work. If that is the case, could the authors describe this a bit further?

---

> ### Author Response · Authors · 2018-01-05
> **We presented our work in the context of the existing literature more carefully and clarified our contributions.**
>
> Thank you for the detailed feedback and understanding our contributions. We significantly revised the manuscript considering the reviewer's concerns. In what follows we list the concerns raised by the reviewer and provide our detailed replies:
>
> -- Comment: The paper provide similar results to the previous work.
> -- Response: We significantly revised the presentation and clarified our contributions. We also used our framework to include additional results. In short, our contributions are summarized as follows:
> • Formally state the local openness property and its use in studying local/global equivalence of optimization problems arising from training non-convex deep models.
> • Provide a complete characterization of the local openness of the matrix multiplication mapping.
> • Show that every local optimum of a two layer linear network optimization problem is globally optimal. Unlike many existing results in the literature, our result requires no assumption on the target data matrix Y , and input data matrix X
> • Develop “almost complete” characterization of the local/global optima equivalence of multi- layer linear neural networks, and provide various counterexamples to show the necessity of each assumption.
> • Show global/local optima equivalence of non-linear deep models having certain pyramidal struc- ture. Unlike some existing works, our result requires no assumption on the differentiability of the activation functions and can go beyond “full-rank cases. In this case, we do agree with the reviewer that we do not allow wide intermediate layers. We explicitly mentioned this in our revised manuscript.
>
> -- Comment: If problem (4) is not considered at all in this paper (in its full generality that considers matrix completion and matrix sensing as special cases), then the authors could just start with the model in (5).
> -- Response: We revised accordingly.
>
> -- Comment: Remark 1 has a nice example - could this example be shown with Y not being the all-zeros vector?
> -- Response: For the given dimensions (m=2, k=1, n=2), it is not possible. The reason is that if both vectors are non-zero, then they are both full rank. Hence, according to our main result on local openness of the matrix product, our mapping is locally open. However, one can easily con- struct other non-zero examples for larger dimensions using our main result as our theorem provides a complete characterization.
>
> --Comment: In section 5, the authors make a connection with the work of Ge et al. 2016. They state that the problems in (10)-(11) constitute generalizations of the symmetric matrix completion case, considered in Ge et al. 2016. However, in that work, the main difficulty of proving global optimality comes from the randomness of the sampling mask operator (which introduces the notion of incoherence and requires results in expectation). It is not clear, and maybe it is an overstatement, that the results in section 5 generalize that work. If that is the case, could the authors describe this a bit further?
> -- Response: Indeed, we only consider the fully-observed matrix completion problem. The matrix com- pletion part has been de-emphasized in the revised manuscript.

---

### Official Review · AnonReviewer3 · 2017-12-04
**Good idea but writing/execution is not upto the mark.**

**Rating:** 5
**Confidence:** 4

**Review:**

Summary:

This paper studies the geometry of linear and neural networks and provides conditions under which the local minima of the loss are global minima for these non-convex problems. The paper studies locally open maps, which preserve the local minima geometry. Hence a local minima of l(F(W)) is a local minima of l(s) when s=F(W) is a locally open map. Theorem 3 provides conditions under which the multiplication X*Y is a locally open map. For a pyramidal feed forward net, if the weights in each layer have full rank,  input X is full rank, and the link function is invertible, then that local minima is a global minima.

Comments:

The locally open maps (Behrends 2017) is an interesting concept. However I am not convinced that the paper is able to show stronger results about the geometry of linear/neural networks. Further the claims all over the paper, comparing with the existing works. are over the top and not justified. I believe the paper needs a significant rewriting.

The results are not a strict improvement over existing works. For neural networks, Nguyen and Hein (2017) assume the link function is differentiable. This paper assumes the link function is invertible. Both papers can handle sigmoid/tanh, but cannot handle ReLU.

Results for linear networks are not an improvement over existing works. Paper claims to remove assumption on Y, but they get much weaker results as they cannot differentiate between saddle points and global minima, for a critical point.  Results are also written in a confusing way as stating each critical point is a saddle or a global minima. Instead the presentation can be simplified by just discussing the equivalency between local minima and global minima, as the proposed framework cannot handle critical points directly.

Proof of Lemma 7 seems to have typos/mistakes. What is \bar{W_i}? Why are the first two equations just showing d_i \leq d_i ? How do you use this to conclude locally openness of \mathcal{M}?

Authors claim their result extends the results for matrix completion from Ge et al. (2016) . This is false claim as (10) is not the matrix completion problem with missing entries, and the results in Ge et al. (2016) do not assume any non-degeneracy conditions on W.

---

> ### Author Response · Authors · 2018-01-05
> **We relaxed many assumptions and show the necessity of the remaining ones for linear networks**
>
> We significantly revised the manuscript considering your comments. In what follows we list the concerns raised by the reviewer and provide our detailed responses:
>
> -- Comment: Paper need significant revisions in terms of comparison with existing results.
> -- Response: We believe the comment was addressed in the revised manuscript. However, we appreciate any new feedback.
>
> -- Comment: Nguyen and Hein (2017) assume the link function is differentiable. This paper assumes the link function is invertible. Both papers can handle sigmoid/tanh, but cannot handle ReLU.
> -- Response: Notice that in the paper by Nguyen and Hein, they also assume strict monotonicity activation (which implies invertibility). Also note that, while our result cannot handle ReLU functions, leaky ReLU activation functions satisfy our assumptions. This has been clarified in the revised manuscript.
>
> -- Comment: Results for linear networks are not an improvement over existing works.
> -- Response: We significantly revised the manuscript to clarify our contributions for linear networks. In short, our contributions for linear networks are the followings:
> • Show that every local optimum of a two layer linear networks is globally optimal. Unlike many existing results in the literature, our result requires no assumption on the target data matrix Y , and input data matrix X.
> • Develop "almost complete characterization" of the local/global optima equivalence of multi-layer linear neural networks, and provide various counterexamples to show the necessity of each assumption.
>
> -- Comment: Proof of Lemma 7 is not clear.
> -- Response: We agree with the reviewer that some parts in the original proof was not clear. Enjoy our revised detailed proof.
>
> -- Comment: The problem considered in the manuscript is the fully observed matrix completion prob- lem and thus the results in the manuscript do not extend the results for matrix completion from Ge et al. (2016). Moreover, the results in Ge et al. (2016) do not assume any non-degeneracy conditions on W.
> -- Response: The matrix completion part has been de-emphasized, and the non-degeneracy condition was relaxed.

---

### Official Review · AnonReviewer2 · 2017-12-06
**Theory building work for linear deep networks, and some nonlinear networks. Redresses and unifies some existing results in a clean way.**

**Rating:** 6
**Confidence:** 4

**Review:**

The paper studies the local optima of certain types of deep networks. It uses the notion of a locally open map to draw equivalences between local optima and global optima. The basic idea is that for fitting nonlinear models with a convex loss, if the mapping from the weights to the outputs is open, then every local optimum in weight space corresponds to a local optimum in output space; by convexity, in output space every local optimum is global.

This is mostly a “theory building” work. With an appropriate fix, lemma 4 gives a cleaner set of assumptions than previous work in the same space (Nguyen + Hein ’17), but yields essentially the same conclusions.

The notion of local openness seems very well adapted to deriving these type of results in a clean manner. The result in Section 3 on local openness of matrix multiplication on its range (which is substantially motivated by Behrends 2017) may be of independent interest. I did not check the proof of this result in detail, but it appears to be correct. For the linear, deep case, the paper corrects imprecisions in the previous work (Lu + Kawaguchi).

For deep nonlinear networks, the results require the “pyramidal” assumption that the dimensionality is nonincreasing with respect to layer and (more restrictively) the feature dimension in the first layer is larger than the number of input points. This seems to differ from typical practice, in the sense that it does not allow for wide intermediate layers. This seems to be a limitation of the methodology: unless I'm missing something, this situation cannot be addressed using locally open maps.



There are some imprecisions in the writing. For example, Lemma 4 is not correct as written — an invertible mapping \sigma is not necessarily locally open. Take $\sigma_k(t) = t for t rational and -t for t irrational$ as an example. This is easy to fix, but not correct as written.

Despite mentioning matrix completion in the introduction and comparing to work of Ge et. al., the paper does not seem to have strong implications for matrix completion. It extends results of Ge and collaborators for the fully observed symmetric case to non-symmetric problems. But the main interest in matrix completion is in the undersampled case — in the full observed case, there is nothing to complete.

---

> ### Author Response · Authors · 2018-01-05
> **We presented our work in the context of the existing literature more carefully and clarified our contributions.**
>
> We would like to thank the reviewer for the careful reading of the manuscript. We significantly revised our submission considering the reviewer's comments. In our revision, we relaxed almost any assumption possible. For example, we relaxed the full rankness of X and Y in the two-layer linear neural networks, and provide an “almost complete” characterization of the local/global optima equivalence of multi-layer linear neural networks. We also included multiple counterexamples to show the necessity of the remaining set of assumptions. To clarify the contributions of the paper, we re-wrote the abstract. In short, our contributions are summarized as follows:
>
> • Formally state the local openness property and its use in studying local/global equivalence of optimization problems arising from training non-convex deep models.
> • Provide a complete characterization of the local openness of the matrix multiplication mapping.
> • Show that every local optimum of a two layer linear network optimization problem is globally optimal. Unlike many existing results in the literature, our result requires no assumption on the target data matrix Y , and input data matrix X
> • Develop "almost complete" characterization of the local/global optima equivalence of multi- layer linear neural networks, and provide various counterexamples to show the necessity of each assumption.
> • Show global/local optima equivalence of non-linear deep models having certain pyramidal struc- ture. Unlike some existing works, our result requires no assumption on the differentiability of the activation functions. In this case, we do agree with the reviewer that we do not allow wide intermediate layers. We explicitly mentioned this in our revised manuscript.
>
> In what follows we list the concerns raised by the reviewer and provide our detailed replies:
>
> --Comment: There are some imprecisions in the writing. For example, Lemma 4 is not correct as written an invertible mapping σ is not necessarily locally open. Take σk(t) = t for t rational and −t for t irrational as an example. This is easy to fix, but not correct as written.
> --Response: Correct. We fixed it in our revision.
>
> -- Comment: Despite mentioning matrix completion in the introduction and comparing to work of Ge et. al., the paper does not seem to have strong implications for matrix completion. It extends results of Ge and collaborators for the fully observed symmetric case to non-symmetric problems. But the main interest in matrix completion is in the under-sampled case in the full observed case, there is nothing to complete.
> -- Response: The matrix completion part has been de-emphasized in the revised manuscript.

---

### Decision · Program_Chairs · 2018-01-29
**ICLR 2018 Conference Acceptance Decision**

**Decision:**

Invite to Workshop Track

**Comment:**

The paper nicely unifies previous results and develops the property of local openness. While interesting, I find the application to multi-layer linear networks extremely limiting. There appears to be a sub-field in theory now focusing on solely multi-layer linear networks which is meaningless in practice. I can appreciate that this could give rise to useful proof techniques and hence, I am recommending it to the workshop track with the hope that it can foster more discussions and help researchers move away from studying multi-layer linear networks.